# Defining Expertise:
# Applications to Treatment Effect Estimation

**Alihan Hüyük***,  **Qiyao Wei***,  **Alicia Curth**,  **Mihaela van der Schaar**
University of Cambridge

## Abstract

Decision-makers are often experts of their domain and take actions based on their domain knowledge. Doctors, for instance, may prescribe treatments by predicting the likely outcome of each available treatment. Actions of an expert thus naturally encode part of their domain knowledge, and can help make inferences within the same domain: Knowing doctors try to prescribe the best treatment for their patients, we can tell treatments prescribed more frequently are likely to be more effective. Yet in machine learning, the fact that most decision-makers are experts is often overlooked, and *"expertise"* is seldom leveraged as an inductive bias. This is especially true for the literature on treatment effect estimation, where often the only assumption made about actions is that of overlap. In this paper, we argue that expertise—particularly the *type of expertise* the decision-makers of a domain are likely to have—can be informative in designing and selecting methods for treatment effect estimation. We formally define two types of expertise, *predictive* and *prognostic*, and demonstrate empirically that: (i) the prominent type of expertise in a domain significantly influences the performance of different methods in treatment effect estimation, and (ii) it is possible to predict the type of expertise present in a dataset, which can provide a quantitative basis for model selection.

## 1 Introduction

Those responsible for making important decisions are often experts of their domain, and they take actions based primarily on domain knowledge. We rely on doctors, for instance, to make treatment decisions for patients. When a patient visits the clinic, their doctor may try to predict the likely outcome of each possible treatment and follow the best treatment route for the patient (Centor, 2007). Another example is teaching: A good teacher would assess the different levels of understanding within their classroom and may choose to tailor their explanations for those who are struggling the most with the lesson (Rosenshine, 2012). When experts take informed actions based on domain knowledge, they naturally impart part of their knowledge to those actions, hence expert actions can in turn be informative in making inferences within the same domain. For instance, knowing that doctors aim to prescribe the best treatment for their patients, we can infer that a treatment prescribed more frequently is likely to be more effective than alternatives. Similarly, considering that teachers focus their explanations on students who need the most help, we can identify more accurately which students might be struggling to understand the lecture. Yet in some fields of machine learning, the fact that most decision-makers tend to be domain experts is often overlooked as a potential source of information. *Expertise* is seldom formalized as an assumption and leveraged as an inductive bias.

Perhaps the most prominent machine learning problem in which the notion of expertise arises quite naturally, while its potential as an inductive bias is usually neglected, is that of personalized *treatment effect estimation*. Estimating the effects of actions, treatments, or interventions is a central concern in numerous domains, and as such, a plethora of machine learning methods has been proposed for estimating treatment effects based on observational data collected by decision-makers (e.g. Bica et al., 2020a; Curth & van der Schaar, 2021a). These methods become susceptible to confounding when those decision-makers happen to be experts and assign treatments based on factors that influence the outcomes of their assignments. As a consequence, treatment effects can generally only be identified if all such confounding factors are recorded in data, and when this is the case, correcting for the resulting shift in covariates across treatment groups is considered a major challenge of the setting (Johansson et al., 2016; Shalit et al., 2017; Assaad et al., 2021). Surprisingly, despite often being one of the principal reasons behind confounding, the expertise of decision-makers is almost never formalized as

---

*Authors contributed equally. Correspondence to: Qiyao Wei <qw281@cam.ac.uk>

an assumption in the literature on treatment effect estimation. Rather, the standard—and often the only–assumption made regarding a decision-maker's policy is that of *overlap*. This assumption states that all treatments must have a non-zero probability of being assigned to any individual so that the resulting data has enough variability to identify treatment effects (Rubin, 2005). As far as expertise is concerned, the overlap assumption means that the decision-maker cannot have perfect knowledge of their domain, allowing them to consistently take the same action given the same situation. Then, by far the most common approach in the machine learning literature for addressing the effects of confounding is to try and remove the imbalances in data created by the decision-maker's policy—for instance, by learning balancing representations according to which the decision-maker's actions appear to be taken randomly with no expertise (Johansson et al., 2016; Shalit et al., 2017; Bica et al., 2019).

In this paper, we argue that methods for treatment effect estimation should consider making more specific assumptions regarding expertise—-beyond just overlap, which only rejects the possibility of *perfect expertise*. Specifically, identifying in what manner the decision-makers of a domain typically exercise their expertise, should inform the design of methods in that domain. To this end, we distinguish two types of expertise: (i) *predictive expertise*, where actions are based on treatment effects only, and (ii) *prognostic expertise*, where actions are based on all potential outcomes more generally.[1] For instance, the doctors in our earlier example happened to have predictive expertise as their goal was to prescribe treatments with the largest benefit, whereas the teachers had prognostic expertise as they aimed to identify students with an overall weaker understanding of the subject, rather than just predicting how much the student could benefit from focused explanations. Our argument, then, is that the prominent type of expertise within a domain matters when designing and selecting models. For instance, we will see in our experiments that balancing representations could be a more suitable approach in domains with predominantly prognostic expertise (e.g. education) than in domains with predominantly predictive expertise (e.g. healthcare). This insight has practical value because model selection is a fundamental issue in treatment effect estimation: The ground-truth treatment effects are almost never observed, which makes conventional model validation infeasible Curth & van der Schaar (2023). Reasoning about expertise thus offers one solution. In our experiments, we will also see that it is possible to estimate the amount of predictive vs. prognostic expertise that the decision-maker collecting a dataset had, which can even provide a *quantitative* basis for model selection.

**Contributions**     • *Conceptually,* we introduce the idea that a decision-maker's policy should not always be considered a nuisance that causes covariate shift, but rather the fact that it often may high expertise could be leveraged as an inductive bias. • *Technically,* we provide a definition of what it means for a policy to have expertise (Sec. 3): Expertise is the extent to which variations in a policy's actions coincide with variations in treatment effects—for predictive expertise—or potential outcomes—for prognostic expertise. We show theoretically that high expertise leads to a greater shift in covariates and poor overlap, making it even more critical to leverage it as an inductive bias. • *Empirically,* we demonstrate that: (i) the type and the amount of expertise present in a dataset significantly influences the performance of different methods for treatment effect estimation (Sec. 4.1), and (ii) it may be possible to classify datasets according to what type of expertise they reflect and thereby identify what methods might be more or less suitable for a given dataset—we propose a pipeline that does this (Sec. 4.2).

## 2 PROBLEM SETUP: TREATMENT EFFECT ESTIMATION

We consider the *standard static setup* for treatment effect estimation (e.g. Curth & van der Schaar, 2021a). This setup assumes a data-generating process where, at each round of decision making, a new subject arrives with features $X \in \mathcal{X}$. These features are distributed according to $\alpha \in \Delta(\mathcal{X})$ such that $X \sim \alpha$, where $\Delta(\mathcal{X})$ is the set of all distributions over $\mathcal{X}$. Each subject is intrinsically associated with two *potential outcomes* (as in Rubin, 2005): $Y_0 \in \mathcal{Y}$ is the baseline outcome that would be realized without treatment and $Y_1 \in \mathcal{Y}$ is the outcome that would be realized if the subject is to be treated. These outcomes are distributed according to $\rho_0 \in \Delta(\mathcal{Y})^{\mathcal{X}}$ and $\rho_1 \in \Delta(\mathcal{Y})^{\mathcal{X}}$ respectively such that $Y_0 \sim \rho_0(X)$ and $Y_1 \sim \rho_1(X)$. After the subject's arrival, a decision-maker assigns them a treatment $A^{\pi} \in \{0, 1\}$, where $A^{\pi} = 0$ denotes no treatment—which we call the "negative" treatment—while $A^{\pi} = 1$ denotes that the subject is treated—which we call the "positive" treatment. These assignments are made according to some policy $\pi \in \Delta(\{0, 1\})^{\mathcal{X}}$ such that $A^{\pi} \sim \pi(X)$. Once a treatment is assigned, the corresponding potential outcome is realized and observed, which we denote as $Y^{\pi} = Y_{A^{\pi}}$.

---

[1]This terminology is inspired by the medical literature; *predictive* biomarkers are informative of the treatment effect and *prognostic* biomarkers are of outcomes regardless of treatment (Ballman, 2015; Sechidis et al., 2018).

**The treatment effect estimation problem** Suppose we are given an observational dataset $\mathcal{D} = \{x^i, a^i, y^i\}_{i=1}^n$ generated by the process we have just described, consisting of subjects $x^i \sim \alpha$, actions (i.e. treatment decisions) $a^i \sim \pi(x^i)$, and factual outcomes $y^i \sim \rho_{a_i}(x_i)$. Then, the treatment effect estimation problem is to determine, based on this observational dataset $\mathcal{D}$, the *conditional average treatment effects* (CATE): $\tau(x) = \mathbb{E}[Y_1 - Y_0 | X = x]$. Generally, this can only be achieved under the *overlap assumption*—that is $\pi(x)[a] > 0$ for all $x \in \mathcal{X}, a \in \{0, 1\}$ (Rubin, 2005). Our setup implicitly assumes *no hidden confounding* as features $\{x^i\}$ are fully observed in $\mathcal{D}$ and $a^i \sim \pi(x^i)$.

## 3 Defining expertise

Before we can provide quantitative evidence in support of our main argument—that is *expertise can act as an inductive bias and inform model selection in treatment effect estimation*—we first need to define formally what we refer to as expertise. This section first states mathematical definitions of prognostic and predictive expertise, and then discusses the motivation behind these definitions (Section 3.1) as well as their implications for the treatment effect estimation problem (Section 3.2). Later in Section 4, we return back to our main argument and present empirical results based on our definitions here.

What we want to capture as *expertise* is to what extent the actions of a decision-maker are informed by how a subject's features shape their potential outcomes—we call this general type of expertise *prognostic expertise*. When a policy $\pi$ has high prognostic expertise, it should be possible to explain variations in its actions $A^\pi$ mostly through the variability of potential outcomes $Y_0, Y_1$—the policy should take different actions under different circumstances only because the potential outcomes are also different. In a similar vein, we aim to capture *predictive expertise*, where actions are *specifically* informed by the treatment effect (rather than the potential outcomes more generally). When a policy $\pi$ has high predictive expertise, variations in $A^\pi$ should mostly be due to the variability of $Y_1 - Y_0$.

As our intuitive understanding of expertise is tightly linked to the variability of actions under different policies, we start by quantifying *action variability* as the entropy of actions $A^\pi$ for a given policy $\pi$:

$$\mathbb{H}[A^\pi] = -\sum\nolimits_{a \in \{0,1\}} \mathbb{P}\{A^\pi = a\} \log_2 \mathbb{P}\{A^\pi = a\} \tag{1}$$

This entropy can be interpreted as a measure of the inherent "randomness" or "uncertainty" actions $A^\pi$ have. Whenever $A^\pi$ is observed, this randomness is removed—or the uncertainty is resolved—hence entropy $\mathbb{H}[A^\pi]$ can also be interpreted as the amount of information carried by the observations of $A^\pi$.

Since expertise has intrinsically to do with how much of this action variability is due to different subjects having different potential outcomes, what we consider next is the entropy of $A^\pi$ conditioned on potential outcomes: $\mathbb{H}[A^\pi | Y_0, Y_1]$. If variations in $A^\pi$ were to be *entirely* due to variations in $Y_0, Y_1$, then $A^\pi$ would appear deterministic conditioned on $Y_0, Y_1$—that is $\mathbb{H}[A^\pi | Y_0, Y_1] = 0$. We want to capture this as the case with maximal prognostic expertise. Conversely, if treatment decisions are not informed by potential outcomes at all, which would be the case with no prognostic expertise, then the action variability would stay the same regardless of whether we condition on $Y_0, Y_1$—that is $\mathbb{H}[A^\pi | Y_0, Y_1] = \mathbb{H}[A^\pi]$. Generalizing these two extremes to a continuum: The higher the expertise is, the less variable actions should appear to be conditioned on a specific pair of potential outcomes, and the lower $\mathbb{H}[A^\pi | Y_0, Y_1]$ should be relative to $\mathbb{H}[A^\pi]$. Hence, we define prognostic expertise as follows:

**Definition 1** (Prognostic expertise). *The prognostic expertise of policy $\pi$ is defined as*

$$E_{prog}^\pi = 1 - \mathbb{H}[A^\pi | Y_0, Y_1] \, / \, \mathbb{H}[A^\pi] \tag{2}$$

Notice that $E_{prog}^\pi \in [0, 1]$, where $E_{prog}^\pi = 1$ for the maximal-expertise case and $E_{prog}^\pi = 0$ for the no-expertise case. Based on our earlier interpretations of entropy, we can conceptualize prognostic expertise in different ways. In those interpretations, actions had an inherent randomness or uncertainty, and if one were to somehow observe both potential outcomes, *a portion* of this randomness would disappear—or *part* of the uncertainty would be resolved. In this hypothetical scenario, prognostic expertise would be the portion of randomness regarding actions that remains—or part of the uncertainty that is still *not* resolved—even after observing both potential outcomes. Alternatively, $E_{prog}^\pi$ is the proportion of information carried by $A^\pi$ that is also informative of $Y_0, Y_1$ (cf. *mutual information*).

Similar to prognostic expertise, we define predictive expertise as the proportion of action variability that is due to variations in the treatment effect $Y_1 - Y_0$ *specifically* (and not just $Y_0, Y_1$ more generally):

**Definition 2** (Predictive expertise). *The predictive expertise of policy $\pi$ is defined as*

$$E_{pred}^\pi = 1 - \mathbb{H}[A^\pi | Y_1 - Y_0] \, / \, \mathbb{H}[A^\pi] \tag{3}$$

Finally, when a policy has unit expertise, meaning any variation in its actions $A^\pi$ is *entirely* due to the variability of potential outcomes (or the treatment effect), we call that policy a *perfect expert*:

**Definition 3** (Perfect prognostic/predictive expert). *Policy $\pi$ is a perfect prognostic expert if $E^\pi_{prog} = 1$. Similarly, policy $\pi$ is a perfect predictive expert if $E^\pi_{pred} = 1$.*

## 3.1 DISCUSSION ON EXPERTISE

**Why two types of expertise?** *Because actions can be informed by outcomes in two distinct ways:* First, treatments can be assigned based on treatment effects (cf. *predictive expertise*). This may happen when the goal is to achieve the best possible outcome for each subject, which is often the case in health-care (e.g. Graham et al., 2007; Caye et al., 2019). Alternatively, treatments can be assigned according to the outcome that subjects would attain regardless of treatment (cf. *prognostic expertise*). This may happen in cases of self-selection—for instance, in education, students who choose to attend optional lectures might be those who already would have had higher grades even if they did not attend those lectures (Kwak et al., 2019)—or when the goal is to equalize outcomes across subjects—for instance, in social planning, institutions that receive the most funding might be the ones that need it the most, and not necessarily the ones that would benefit the most (Ladd & Yinger, 1994; Betts & Roemer, 1999). In our experiments, we will evaluate the performance of different treatment effect estimation methods for treatment effect estimation under varying amounts of both expertise types (see Figure 4 in Section 4.1).

**Are the two types of expertise related?** *Yes, predictive expertise implies prognostic expertise.* This is because the treatment effect $Y_1 - Y_0$ is a function of potential outcomes $Y_0, Y_1$, hence actions informed by $Y_1 - Y_0$ are also indirectly informed by $Y_0, Y_1$. However, the converse is not necessarily true: A policy might have prognostic expertise but completely lack predictive expertise at the same time.

**Can expertise be measured?** *Not directly*—both types of expertise are *oracle measures*, meaning they cannot be computed in practice as only one of the potential outcomes would be observed. But, they can be estimated via models as we demonstrate with experiments (see Figure 5 in Section 4.2).

**How does expertise differ from optimality?** Optimality is tied to a specific success measure—as in utility functions in economics (Kapteyn, 1985) or reward functions in control/reinforcement learning (e.g. Sutton & Barto, 2018; Holt et al., 2023; 2024; Sun et al., 2024)—and entails high performance w.r.t. that success measure. In contrast, expertise expresses the idea that any variability in treatment assignments is solely motivated by what the potential outcomes (or the treatment effect) could be—independent of any particular success measure. Dependence on a single success measure can be problematic: Policies with factually incorrect information (lacking expertise) might lead to high performance in some measures by coincidence. Conversely, actual experts might appear to be performing poorly according to one measure only because they are trying to optimize another measure.

As an illustration of such cases, consider an environment with features $X = (X_A, X_B, X_C)$ such that $X_i \sim \mathcal{U}(\{-1, 0, 1\})$ for all $i \in \{A, B, C\}$, where $\mathcal{U}(\mathcal{S})$ is the uniform distribution over set $\mathcal{S}$, and outcomes $Y_0 = 0$ and $Y_1 = X_A + X_B$ so that the treatment effect is $\tau(x) = x_A + x_B$. One goal in this environment could be to maximize realized outcomes with success measure $\mathbb{E}[Y^\pi]$. Then, consider two policies: (i) a *misinformed* policy $\pi_{mis}(x) = \mathbb{1}\{x_A + x_C > 0\}$ which tries to maximize outcomes by assigning the treatment whenever its effect is positive, but has incorrectly identified an irrelevant variable $X_C$ in place of $X_B$, and (ii) a *risk-averse* policy $\pi_{risk}(x) = \mathbb{1}\{x_A + x_B > 1\}$ which has correct information, but instead of maximizing outcomes, tries to avoid adverse negative outcomes hence assigns the treatment only when its effect is overwhelmingly positive. In this scenario, the misinformed policy still happens to perform better than the risk-averse policy in maximizing outcomes: $\mathbb{E}[Y^{\pi_{mis}}] > \mathbb{E}[Y^{\pi_{risk}}]$. However, its actions are varied based on variables that should ideally not be a consideration in decision-making, and its performance comes at the expense of individuals with features $x$ who would have benefited from the treatment as $\tau(x) > 0$ but did not receive it as $x_C \ll 0$. Appropriately, the (predictive) expertise of the risk-averse policy happens to be higher, $E^{\pi_{risk}}_{pred} > E^{\pi_{mis}}_{pred}$.

**Are experts more desirable than optimal policies?** *Sometimes yes, for instance in healthcare!* Since treatments might have different effects for different patients, variability is needed in which treatments are prescribed to which patients. However, this variability should occur only with respect to variables that are clinically relevant and not with respect to arbitrary characteristics. In this context, policies like $\pi_{risk}$ could be more desirable: It might fall short of maximizing outcomes but varies its actions only when needed. Policies like $\pi_{mis}$, on the other hand, possibly have *unwanted variation* where subjects are treated preferentially based on factors that should not be relevant to the outcome of interest.

### 3.2 IMPLICATIONS OF EXPERTISE FOR TREATMENT EFFECT ESTIMATION

Before we move onto investigating, practically, how expertise can act as an inductive bias and inform model selection in treatment effect estimation, it would be insightful to first understand the implications of having high expertise for the treatment effect estimation problem itself. Below, we show that high expertise necessarily leads to poor overlap, which overall makes the problem more challenging.

Characterizing this mathematically requires us to first differentiate between two distinct sources of action variability. To that end, suppose $X \sim \mathcal{B}(1/2)$, where $\mathcal{B}(p)$ is the Bernoulli distribution with success probability $p$, and consider: (i) a *uniform policy* $\pi_{unif}(x)[a] = 1/2$, which assigns treatments uniformly at random for all subjects, and (ii) a *preferential policy* $\pi_{pref}(x)[a] = \mathbb{1}\{a = x\}$, which assigns half of the subjects one treatment and the other half the other treatment. Both policies have the same action variability, $\mathbb{H}[A^{\pi_{unif}}] = \mathbb{H}[A^{\pi_{pref}}] = 1$. However, while the uniform policy is highly stochastic, the preferential policy behaves in a completely deterministic way given a subject $x \in \mathcal{X}$. We capture the difference between these two policies by defining the notion of *in-context action variability*:

**Definition 4** (In-context action variability). *The in-context action variability of policy $\pi$ is defined as*

$$C^{\pi} = \mathbb{H}[A^{\pi}|X] \,/\, \mathbb{H}[A^{\pi}] \tag{4}$$

Notice that $C^{\pi} \in [0, 1]$, and the uniform policy has unit in-context variability, $C^{\pi_{unif}} = 1$, while the preferential policy has zero in-context variability, $C^{\pi_{pref}} = 0$. In-context action variability is useful to define because it is related directly to the *overlap assumption*. For it to hold, the in-context variability has to be non-zero, $C^{\pi} = 0$ would imply that the overlap assumption is violated. On the other extreme, when treatments are assigned uniformly at random (as in a randomized controlled trial), the in-context variability would reach its maximal value. As such, in-context action variability can be thought of more generally as a measure of the amount of overlap the decision-maker's policy has or how "difficult" it is to estimate treatment effects from data collected by a particular policy. Keeping this interpretation in mind, what we show next is that the total expertise and in-context action variability of a policy is bounded—meaning high expertise would lead to low in-context action variability hence poor overlap:[2]

**Proposition 1** (Boundedness of expertise and in-context action variability). *For all $\pi \in \Delta(\{0,1\})^{\mathcal{X}}$,*

$$E^{\pi}_{prog} + C^{\pi} \leq 1 \quad and \quad E^{\pi}_{pred} + C^{\pi} \leq 1 \tag{5}$$

*Proof.* Appendix C. □

This is why leveraging expertise as an inductive bias is so crucial: Cases where more information can be gained through expertise (i.e. cases with high expertise) happen to align with cases where treatment effect estimation is particularly hard due to poor overlap (Figure 1). According to one extreme of Proposition 1, any perfect expert—predictive or prognostic–must be making deterministic decisions with respect to subject features—violating the overlap assumption necessary for non-parametric treatment effect estimation:

**Proposition 2** (Determinism of perfect experts). *If policy $\pi$ is a perfect expert—that is either $E^{\pi}_{prog} = 1$ or $E^{\pi}_{pred} = 1$—then $C^{\pi} = 0$.*

Now, a natural question one might ask next is: *Should we still care about treatment effect estimation when the decision-maker's policy is already a perfect expert?* Possibly "Yes!" for two reasons:

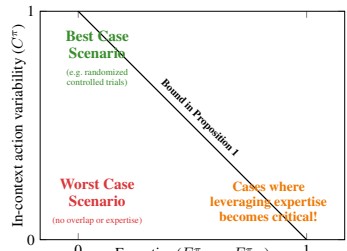

**Figure 1:** The higher the expertise of a policy, the lower its in-context action variability, hence its overlap, has to be (Prop. 1). When expertise is high, leveraging it becomes critical as overlap would be low, making CATE estimation more challenging.

(i) As we have stressed earlier, being an expert does not necessarily imply optimal performance with respect to any arbitrary success measure. In particular, a decision-maker might be a perfect expert because they are optimizing one success measure, while estimating potential outcomes or the treatment effect accurately might allow us to optimize another success measure of interest.

(ii) Even if an expert were to be readily optimal with respect to a desired objective, one might still be interested in formalizing their policy in mathematical terms—rather than being reliant on the internal thought process of a human—thereby extracting the knowledge that might otherwise only be known (or intuitive) to the expert. In fact, this happens to be the main goal of *inverse decision modeling* (Bica et al., 2020b; Qin et al., 2021; Jarrett et al., 2021; Hüyük et al., 2021; 2022a;b).

---

[2]In this paper, we continue to assume overlap and only ever consider policies with *high expertise* rather than *perfect expertise*—having poor overlap does not mean that the overlap assumption is violated.

## 4 APPLICATIONS TO TREATMENT EFFECT ESTIMATION

Having established the necessary language for discussing expertise (and action variability) in a quantitative manner, we can finally return to our main practical argument: *Expertise—particularly the type of expertise present in a dataset—can act as an inductive bias and inform the selection of methods for treatment effect estimation—policies should not necessarily be viewed as nuisances only (as in balancing representations).* This section provides two empirical demonstrations in support of our argument:

(i) In Section 4.1, we analyze the operating characteristics of various treatment effect estimation methods under different expertise-related scenarios. Our analysis reveals an important insight: Learning balancing representations can indeed hurt performance in treatment effect estimation *when decision-makers have predictive expertise*. Under *purely prognostic* expertise however (without any predictive expertise), balancing representations can select against features that affect outcomes despite being unrelated to the treatment effect—thereby improving performance.

(ii) In Section 4.2, we estimate the amount of predictive vs. prognostic expertise a decision-maker has, which not only enables one to evaluate the expertise of different decision-makers, but also to predict which treatment effect estimation methods might perform better for a given dataset.

**Simulation environment** As is common in the treatment effect estimation literature, we need to construct a synthetic data-generating mechanism to ensure that potential outcomes are *known by design* (Curth et al., 2021; Hüyük et al., 2024). Inspired by the simulator in Crabbé et al. (2022), and similar to them, we start with covariates $X \in \mathbb{R}^d$ from real-world datasets. We designate each covariate as either being *prognostic*, *predictive*, or *irrelevant* such that $X = (X_{prog}, X_{pred}, X_{irr}) \in \mathbb{R}^{d_{prog} \times d_{pred} \times d_{irr}}$. Then, we generate potential outcomes such that $Y_a = \langle w_{prog}, X_{prog} \rangle + \langle w_a, X_{pred} \rangle + \eta_a$, where $w_{prog} \in \mathbb{R}^{d_{prog}}$ and $w_0, w_1 \in \mathbb{R}^{d_{pred}}$ are weights with components sampled independently from the uniform distribution over $[-1, 1]$, and $\eta_0, \eta_1$ are noise variables sampled from the normal distribution with $\mu = 0$ and $\sigma = 0.1$. According to this process: (i) the treatment effect $\tau(x) = \langle w_1 - w_0, x_{pred} \rangle$ depends only on predictive variables and not on prognostic variables, and (ii) neither of the two potential outcomes depend on irrelevant variables. (A full description of our simulator can be found in Appendix E.)

**Decision-making policies** Using Figure 1 as a map, we vary the expertise of policies in this environment on two distinct axes (Figure 2): (i) from "best case scenario" to the high-expertise scenario (**Best→Expert**), and (ii) from "worst case scenario" to the high-expertise scenario (**Worst→Expert**). Consider the (soft) predictive expert $\pi_{soft}$ that is more likely to assign the positive treatment if the treatment effect is positive: $\pi_{soft}(x)[1] = \sigma(\tau(x)/\beta)$, where $\sigma(x) = 1/(1 + e^{-x})$ is the sigmoid function and the temperature parameter $\beta \in \mathbb{R}_+$ controls the in-context action variability. Then:

**Figure 2:** During our simulations, we increase the expertise: (i) by decreasing $\beta$ in $\pi_{soft}$ (**Best→Expert**), which also decreases the in-context action variability (i.e. the overlap), and (ii) by decreasing $d$ in $\pi_{mis}$ (**Worst→Expert**, as seen in Fig. 1).

(i) For *Best→Expert*, we simply vary the parameter $\beta$. At the limit $\beta \to \infty$, $\pi_{soft}$ becomes equivalent to the random policy $\pi_{rand}(x)[a] = 1/2$ (i.e. the "best case scenario"). As $\beta$ gets smaller and smaller, $\pi_{soft}$ gains more and more expertise.

(ii) For *Worst→Expert*, we fix $\beta = 1/4$—and with it also the in-context action variability—but we slowly replace the predictive variables in $X_{pred}$ that an expert would take into consideration with irrelevant variables from $X_{irr}$ instead. Letting $d \in \{0, \ldots, \min\{d_{pred}, d_{irr}\}\}$ be the number of predictive variables that are replaced, we construct *misspecified policies* $\pi_{mis}(x) = \sigma(\langle w_1 - w_0, x'_{pred} \rangle/\beta)$ where $x'_{pred} = ((x_{irr})_{1:d}, (x_{pred})_{d+1:d_{pred}})$. Note that $\pi_{mis}$ becomes equivalent to $\pi_{soft}$ when $d = 0$.

We also consider policies without any predictive expertise that take actions based on a combination of prognostic and irrelevant variables: $\pi_{non\text{-}pred}(x)[1] = \sigma(\langle w_{prog}, x'_{prog} \rangle/\beta)$ where $x'_{prog} = ((x_{irr})_{1:d}, (x_{prog})_{d+1:d_{prog}})$. Similar to before, for *Best→Expert*, we fix $d = \min\{d_{prog}, d_{irr}\}/2$ and vary $\beta$, and for *Worst→Expert*, we fix $\beta = 1/4$ and vary $d$. Despite lacking predictive expertise, $\pi_{non\text{-}pred}$ has prognostic expertise (as long as $d \neq d_{prog}$), hence why we will refer to this case as the *prognostic setting*.

**Benchmark algorithms** Below, we review the existing strategies for treatment effect estimation and consider them *through an expertise lens*. In particular, we highlight the inductive biases regarding expertise that are implicitly encoded in different methods, which as we will discuss, arise as a by-product of unrelated design motivations. (A full description of all algorithms can be found in Appendix F.)

- *Potential outcome predictors:* The methods estimate CATE through prediction of the negative and the positive outcomes separately. Many popular CATE estimators in the machine learning literature (e.g. Johansson et al., 2016; Hassanpour & Greiner, 2019; Künzel et al., 2019; Curth & van der Schaar, 2021b) fall into this class. In our experiments, we consider TARNet of Shalit et al. (2017) as a representative of this class. TARNet first learns a shared representation $\phi : \mathcal{X} \to \mathcal{R}$, which is then taken as input by two separate output prediction heads $f : \mathcal{R} \times \{0, 1\} \to \mathcal{Y}$. Finally, the treatment effect is estimated as $\hat{\tau}(x) = f(\phi(x), 1) - f(\phi(x), 0)$. By focussing on potential outcome prediction *only*, these methods are essentially *agnostic* to any specialized structure that the treatment effect estimation problem has, including policies and any relationship they might have with the potential outcomes (or the treatment effect) such as expertise. Being a neutral method in terms of how policies are treated, we use TARNet as our baseline method ("**Baseline**").

- *Propensity-weighted predictors:* One potential issue that potential outcome predictors are ignorant of is the possibility of *covariate shift*—that is the distribution of covariates in the training set for each potential outcome predictor, $f(\phi(x), 0)$ and $f(\phi(x), 1)$, not being equal to the marginal distribution $\alpha$ of features $X$. One straightforward correction for covariate shift is importance weighting, which in the CATE estimation case is also known as inverse propensity weighting (IPW, e.g. Wager & Athey, 2018; Hassanpour & Greiner, 2019; Assaad et al., 2021; Dorn & Guo, 2022). IPW can be applied to the same architecture as TARNet (i.e. *Baseline*) ("**Propensity**"). In TARNet, functions $\phi, f$ are trained according to some loss function $\mathcal{L} = \sum_i \|y^i - f(\phi(x^i), a^i)\|$, where $\| \cdot \|$ denotes an arbitrary distance metric. IPW corrects for shifts in covariates by re-weighting the contribution of individual data points to this loss function: $\mathcal{L}' = \sum_i \|y^i - f(\phi(x^i), a^i)\| / \pi(x^i)[a^i]$. Now notice that IPW requires policy $\pi$ to be known (or estimated) beforehand, and given a policy $\pi$, it does not leverage any relationship between $\pi$ and the potential outcomes $Y_0, Y_1$. In other words, IPW is indifferent to any expertise the decision-maker's policy might have (but corrects for the possible effects of covariate shift). However, there is one caveat: IPW is extremely sensitive to the stochasticity of policy $\pi$ since propensity scores $\pi(x)[a]$ appear in denominators—more deterministic policies with smaller propensity scores for certain actions lead to estimates with higher variance (Cortes et al., 2010). Having high expertise might mean policies with less in-context variability in their actions (according to Proposition 1) hence less stable estimates when IPW is used.

- *Balancing representations:* While IPW requires a policy to be fully specified beforehand and ignores any relationship between that policy and the potential outcomes, balancing representations (e.g. Johansson et al., 2016; Yao et al., 2018; Bica et al., 2019; Du et al., 2021; Huang et al., 2022) actively try to disentangle the decision-maker's policy and the estimates for potential outcomes from each other. For instance, CFRNet proposed by Shalit et al. (2017) ("**Balancing**") uses the same architecture as TARNet, except that the bias caused by policy $\pi$ is addressed by training function $\phi$ to generate representations that are predictive of potential outcomes $Y_0, Y_1$ but *not* of actions $A^\pi$, so that actions $a^i$ appear to be random with respect to learned representations $z^i = \phi(x^i)$. A typical loss function for learning such representations would be $\mathcal{L} = \sum_i \|y^i - f(z^i, a^i)\| + \|\{z_i\}_{i:a^i=0} - \{z_i\}_{i:a^i=1}\|$. This approach originates from domain adaptation (Ganin et al., 2016), where it was proposed to avoid the variability issues of importance-weighted estimators. In our experiments however, we will see that, when policies have *predictive* expertise, balancing representations actively remove the information carried by that expertise, which is not always desirable.

- *Action-predictive representations:* Finally, in stark contrast with balancing representations, we consider a final strategy that learns function $\phi$ so that representations $z^i = \phi(x^i)$ are actually predictive of actions $a^i$. This strategy encodes *predictive* expertise as an inductive bias in the sense that it assumes that policies and outcome predictors can be represented in a joint space $\mathcal{R}$ from which it is easier to learn function $f$ then from the original space. Such a strategy is implemented as DragonNet in Shi et al. (2019) ("**Action-predictive**")—albeit motivated from a different angle.[3] The loss function for DragonNet looks like $\mathcal{L} = \sum_i (\|y^i - f(z^i, a^i)\| + \log g(z_i)[a_i])$ where function $g : \mathcal{R} \to \Delta(\{0, 1\})$ is trained jointly with functions $\phi, f$ to predict action distributions.

**Performance metric** As our main metric of performance, we consider *precision in estimation of heterogeneous effects* (PEHE)—that is $\mathbb{E}_{X \sim \alpha}[(\hat{\tau}(X) - \tau(X))^2]^{1/2}$ for an estimator $\hat{\tau}(x)$. In the main paper, we focus on a single simulation environment, additional experiments can be found in Appendix D.

---

[3]Shi et al. (2019) consider *average* treatment effect estimation—that is the estimation of $\mathbb{E}[Y_1 - Y_0]$. In this context, policy $\pi$ plays a different, special, role: It is sufficient for adjustment (Rosenbaum & Rubin, 1983), which is what Shi et al. (2019) propose to exploit. Note that $\pi$ is *not* sufficient for adjustment in CATE estimation unless $\mu_a(x) \doteq \mathbb{E}[Y_a | X = x]$ is a function of $\pi(x)$ alone.

## 4.1 PERFORMANCE UNDER DIFFERENT EXPERTISE SCENARIOS

When varying policy $\pi$ as in Figure 2, as we move away from the "best case scenario" in Figure 1 to the case with high expertise and eventually to the "worst case scenario" (*Best→Expert→Worst*), the treatment effect estimation problem gets generally harder—see Figure 3. In order to compensate for this inherent change in the difficulty of estimating treatment effects, in this subsection, we measure the performance of all methods relative to *Baseline*.

Figures 4a and 4b show the PEHE improvement of different methods over *Baseline* for the setting with predictive expertise. Three observations stand out: First, *Action-predictive* achieves better and better performance over *Baseline* as the decision-maker's expertise increases. This is because *Action-predictive* learns variables that are predictive of actions, and when the expertise is high, these variables happen to be good predictors of the treatment effect as well. Second, the performance of *Balancing* degrades relative to *Baseline* as the expertise increases. This is because more relevant

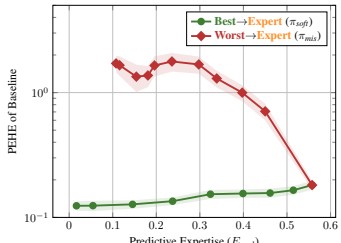

**Figure 3:** As **Best→Expert** (i.e. away from the "best case scenario"), treatment effect estimation gets generally harder and the performance of *Baseline* degrades. Similarly, as **Worst→Expert** (i.e. away from the "worst case scenario") the performance of *Baseline* improves instead.

features becoming predictive of the actions forces *Balancing* to exclude more of those features from its representation space.[4] Third, the relative performance of *Propensity* appears to degrade more consistently as *Best→Expert* than *Worst→Expert*. This supports our earlier intuition that *Propensity* is mostly sensitive to in-context action variability (and overlap) rather than expertise directly—remember that *Best→Expert* reduces the in-context action variability whereas *Worst→Expert* does not (Figure 2).

Figures 4c and 4d show the PEHE improvement of the same methods but for the prognostic setting. In general, we see that the performance of *Action-predictive* becomes worse than *Baseline*. For *Action-predictive*, learning variables that influence actions the most, which happen to be prognostic or irrelevant variables, is no longer directly informative in estimating the treatment effect (but can still be informative in predicting potential outcomes). For *Balancing* on the other hand, having non-predictive variables—especially irrelevant variables—determine actions, most of the time, helps

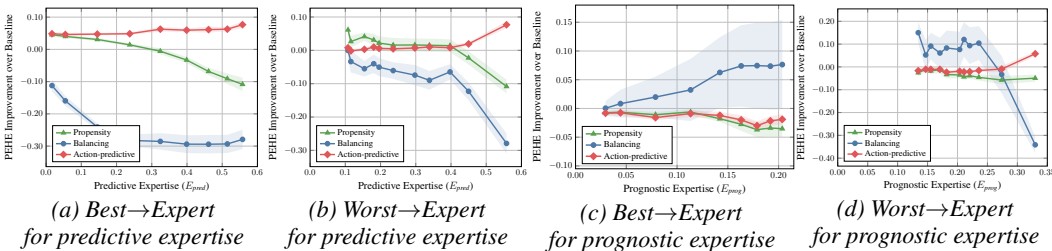

|(a) Best→Expert for predictive expertise|(b) Worst→Expert for predictive expertise|(c) Best→Expert for prognostic expertise|(d) Worst→Expert for prognostic expertise|

**Figure 4:** *Performance improvements over Baseline.* For predictive expertise, *Action-predictive* is able to improve more and more upon *Baseline* by exploiting the increasing expertise (both as *Best→Expert* and *Worst→Expert*). In contrast, *Balancing* gets worse with increasing expertise since the information it discards about the policy becomes more correlated with the treatment effects. However, for prognostic expertise, we observe that having non-predictive variables determine actions can help *Balancing* select against those variables when forming representations, improving performance upon *Baseline*—in most configurations as opposed to *Action-predictive*.

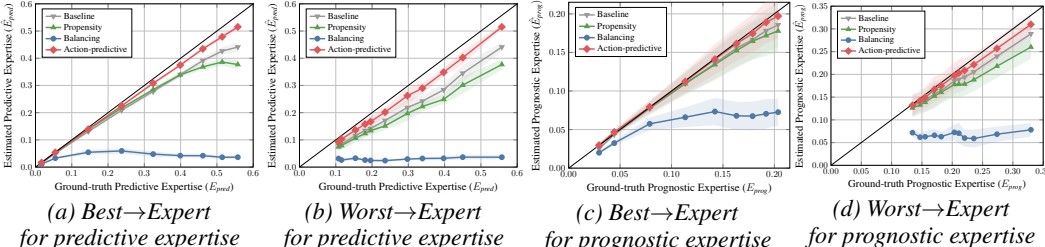

|(a) Best→Expert for predictive expertise|(b) Worst→Expert for predictive expertise|(c) Best→Expert for prognostic expertise|(d) Worst→Expert for prognostic expertise|

**Figure 5:** *Estimated predictive/prognostic expertise.* *Balancing* fails completely since, by design, it removes all information that would have been predictive of expertise. *Propensity* is more sensitive to changes in expertise as *Best→Expert* than *Worst→Expert* (especially for predictive expertise) since the in-context action variability—hence the magnitude of propensity scores—changes as *Best→Expert* while it stays the same as *Worst→Expert*.

[4]This is related to *information loss* in representations as analyzed theoretically in Johansson et al. (2019) for domain adaptation. Their results can potentially be adapted to our setting to provide generalization bounds.

*Balancing* eliminates those variables from its representations, which regularizes the estimation of treatment effects. However, *Balancing* still suffers if the policy depends *only* on prognostic variables.

Finally, perhaps the most important insight we gain from results in Figure 4 is that, on average, different methods perform the best under predictive vs. prognostic expertise. While *Action-predictive* may be more suitable for domains with high predictive expertise, such as healthcare, *Balancing* may be more suitable for domains with high prognostic (and importantly low predictive) expertise, such as education. In the next section, we demonstrate that the prominent type of expertise present in a dataset can be identified by estimating the relative amounts of predictive and prognostic expertise reflected in that dataset with one of our benchmark algorithms. This provides a quantitative basis for deciding whether to use *Action-predictive* or *Balancing* when estimating treatment effects given a particular dataset.

## 4.2 ESTIMATING EXPERTISE

Figure 5 shows that, although expertise is an oracle metric—meaning it cannot be computed directly using observable data—it can be estimated by plugging potential outcome predictions, which all of our methods learn, into (2) and (3). Specifically, given a partition of the outcome space $\mathcal{Y} = \mathcal{Y}_1 \cup \cdots \cup \mathcal{Y}_k$, the predictive expertise can be estimated by treating outcome predictions $\{\hat{y}_a^i\}$ as discrete variables:

$$\hat{E}_{pred} = 1 - \sum_{\substack{a \in \{0,1\} \\ j \in [k]}} \frac{|i:a^i=a, \hat{y}_1^i - \hat{y}_0^i \in \mathcal{Y}_j|}{n} \log_2 \frac{|i:a^i=a, \hat{y}_1^i - \hat{y}_0^i \in \mathcal{Y}_j|}{|i:\hat{y}_1^i - \hat{y}_0^i \in \mathcal{Y}_j|} \bigg/ \sum_{a \in \{0,1\}} \frac{|i:a^i=a|}{n} \log_2 \frac{|i:a^i=a|}{n} \quad (6)$$

Prognostic expertise can be estimated in a similar manner as well. All methods except *Balancing* perform well in this task. This is no surprise as one of the training goals of *Balancing* is to remove all information regarding the decision-maker's policy, including any indicators of its expertise. We see once again that *Propensity* is more sensitive to changes in expertise as *Best→Expert* than *Worst→Expert* (since *Best→Expert* affects the magnitude of propensity scores whereas *Worst→Expert* does not).

Notice that *Action-predictive* performs the best in estimating expertise universally across all configurations, hence it can be used to identify whether a dataset has predominantly predictive expertise or prognostic expertise, and according to the identified expertise type, choose between *Action-predictive* or *Balancing* for estimating treatment effects. Given a dataset $\mathcal{D}$, we first estimate both $E_{pred}$ and $E_{prog}$ using *Action-predictive*. Then, we determine which treatment effect estimation method to use based on the ratio $\hat{E}_{pred}/\hat{E}_{prog}$: If this ratio is larger than a threshold of $1/2$, we use *Action-predictive* for treatment effect estimation, otherwise we use *Balancing*. We call this pipeline "**Expertise-informed**" (Figure 6). Table 1 reports the performance of all our methods, including this new pipeline, averaged across the datasets generated by all the policies we have considered so far— that is $\pi_{soft/mis}$ and $\pi_{non-pred}$ with varying $\{\beta, d\}$ as *Best→Expert* and *Worst→Expert*. We see that, by accurately identifying the prominent type of expertise present in a dataset, and selecting the most suitable method between *Action-predictive*

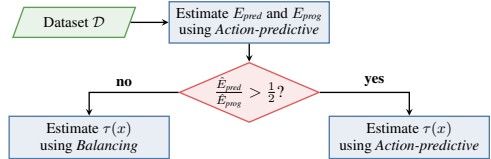

**Figure 6:** *Flow diagram of Expertise-informed.* First, the dominant type of expertise present in dataset $\mathcal{D}$ is identified using *Action-predictive*. Then, a suitable method for CATE estimation is selected accordingly.

**Table 1:** *PEHE of various methods averaged across predictive, prognostic, and all datasets.* By accurately identifying the type of expertise present in datasets, and then selecting between *Action-predictive* and *Balancing* for CATE estimation accordingly, *Expertise-informed* achieves the best-of-both-worlds performance.

| Method | Predictive Datasets | Prognostic Datasets | All Datasets |
|---|---|---|---|
| Baseline | 0.784 (0.130) | 1.483 (0.258) | 1.134 (0.194) |
| Propensity | 0.786 (0.126) | 1.511 (0.259) | 1.149 (0.193) |
| Balancing | 0.936 (0.131) | **1.439 (0.243)** | 1.188 (0.187) |
| Action-predictive | **0.751 (0.128)** | 1.495 (0.259) | 1.123 (0.194) |
| Expertise-informed | **0.751 (0.128)** | **1.439 (0.243)** | **1.096 (0.185)** |

and *Balancing* for that dataset, *Expertise-informed* achieves the best-of-both-worlds performance.

## 5 CONCLUSION

In this paper, we introduced a technical language that defines what expertise should be conceptualized as in treatment effect estimation. Using this language, we have provided an initial demonstration of an important phenomenon: Encoding the intuition that decision-makers are usually experts of their domain—particularly that they often have a certain type of expertise—can act as an inductive bias in estimating treatment effects. In our experiments, we were able to identify what type of expertise is present in a given dataset and select the most suitable treatment effect estimation method for that dataset accordingly. We hope that the definitions we have provided, along with our empirical demonstrations, will encourage future research to capture the same intuition with new approaches.

## REPRODUCIBILITY STATEMENT

We have described the details of our experimental setup in Appendix E for the simulation environment and in Appendix F for the benchmark algorithms. Moreover, the code for reproducing our main experimental results can be found at `https://github.com/QiyaoWei/Expertise` and `https://github.com/vanderschaarlab/Expertise`. We have provided rigorous proofs in Appendix C of Propositions 1 and 2 (as well as the additional Propositions 3 and 4 that appear in Appendix B), which refer to the assumptions we have made in the main paper as needed.

## ACKNOWLEDGMENTS

We would like to thank Ioana Bica, Krzysztof Kacprzyk, Kasia Kobalczyk, Max Ruiz Luyten, Julianna Piskorz, and Andrew Rashbass for their comments, suggestions, and generous feedback. AH, QW and AC gratefully acknowledge funding from the United States Office of Naval Research (ONR), GlaxoSmithKline, and AstraZeneca respectively.

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

## A  FURTHER DISCUSSION

**Why entropy over statistical measures?**    Our definitions of expertise are based on the concept of entropy. An alternative approach could have been to define expertise using statistical measures such as variance. Although variance, similar to entropy, is also indicative of how "random" a random variable is, the two quantities measure fundamentally different things: Variance quantifies the amount of spread around a mean while entropy quantifies uncertainty. When defining expertise, our aim was to capture the uncertainty of outcomes when the actions of a decision-maker are known vs. unknown to an outside observe (the bigger the difference between the two cases, we say the larger the expertise is). Hence, a more direct measure of uncertainty was naturally more suitable to our aim and use case.

On a more technical level, attempting to define expertise through variance would have had two immediate shortcomings:

(i) First, variance is not well defined for categorical variables, such as binary actions in our work, unless we assign numerical values to each category. Even if we were to assign such numeric values (for instance, $A^\pi = 0$ for the negative treatment and $A^\pi = 1$ for the positive treatment), the resulting variance would be sensitive to our assignments (for instance, the variance of $A^\pi$ would have increased if we were to represent the negative treatment as $A^\pi = -1$ instead of $A^\pi = 0$, which should not be a meaningful change as far as expertise is concerned).

(ii) Variance depends on the scale of variables whereas entropy does not. For instance, the variance of $2Y$ would be double the variance of $Y$, while their entropies would be the same: $\mathbb{H}[Y] = \mathbb{H}[2Y]$. Such sensitivity to scale is undesirable when defining expertise—the fact that outcomes are recorded twice as large in a dataset (maybe due to a change of units) should have no effect on expertise.

**Why not use the graphical framework of Pearl?**    We refrained from using the graphical framework of Pearl (2009) when defining expertise because it is specifically not well-equipped to distinguish between prognostic and predictive variables. This is because directional acyclic graphs (DAGs) are notoriously bad at representing effect modification (i.e. predictive variables) natively. There have been proposals to extend the graphical framework to better depict effect modifiers Weinberg (2007), but to the best of our knowledge, no solution has been well established in the literature to this data.

**Expertise in terms of mutual information**    It should be mentioned that the expertise definitions we arrived at in Section 3 happen to be exactly equal to the *mutual information* between actions and outcomes: For predictive expertise, $E_{pred}^\pi = I(A^\pi; Y_1 - Y_0)/\mathbb{H}[A^\pi]$, and for prognostic expertise, $E_{prog}^\pi = I(A^{\pi; Y_0, Y_1})/\mathbb{H}[A^\pi]$.

**Expertise in experimental data**    Experimental data constitute a noteworthy extreme when viewed through the lens of expertise. Particularly in a randomized controlled trial, when the propensity scores are uniform across treatment, there would be no expertise, and appropriately, both the predictive and the prognostic expertise would be equal to zero for datasets collected through a randomized controlled trial (this can be inferred from Proposition 1, a uniform policy $\pi_{unif}$ would have $C^{\pi_{unif}} = 1$ hence $E_{pred}^{\pi_{unif}} = 0$ and $E_{prog}^{\pi_{unif}} = 0$). Consequently, expertise as an inductive bias would of course be less helpful in estimating treatment effects using such datasets, however in that case, the datasets would already be ideal for treatment effect estimation with no confounding bias (the best case scenario in Figure 1). This is why we mainly focused on the high expertise, low overlap setting (the amber region in Figure 1).

While taking advantage of expertise would not be possible for datasets collected through a randomized controlled trial, estimating expertise (as in Section 4.2) can still act as a data-driven way to determine whether the data we have is effectively randomized or not: The closer both $E_{pred}$ and $E_{prog}$ are to $0$, the closer the data would be to trial data, in which case we might prefer conventional supervised learning methods over algorithms specialized for treatment effect estimation.

## B  THEORY OF EXPERTISE

We have mentioned two fairly straightforward facts regarding expertise without formally stating and proving them: (i) the fact that predictive expertise implies prognostic expertise (in Section 3.1), and (ii) the fact that having zero in-context action variability implies that the overlap assumption is violated (in Section 3.2). In this section, we formally state these facts as Proposition 3 and Proposition 4 respectively with accompanying formal proofs in Appendix C.

**Proposition 3.** $E_{pred}^{\pi} \leq E_{prog}^{\pi}$ *for all* $\pi \in \Delta(\{0,1\})^{\mathcal{X}}$.

**Proposition 4.** $C^{\pi} = 0$ *implies that* $\pi(x)[a] = 0$ *for some* $x \in \mathcal{X}, a \in \{0,1\}$.

## C  PROOFS OF PROPOSITIONS

**Proof of Proposition 1**    We unify the proof by letting $Z$ denote $(Y_0, Y_1)$ in the case of prognostic expertise and $Y_1 - Y_0$ in the case of predictive expertise. Then, for both types of expertise, we have

$$E^{\pi} + C^{\pi} = 1 - \mathbb{H}[A^{\pi}|Z] \,/\, \mathbb{H}[A^{\pi}] + \mathbb{H}[A^{\pi}|X] \,/\, \mathbb{H}[A^{\pi}] \tag{7}$$
$$= 1 - \mathbb{H}[A^{\pi}|Z] \,/\, \mathbb{H}[A^{\pi}] + \mathbb{H}[A^{\pi}|X,Z] \,/\, \mathbb{H}[A^{\pi}] \tag{8}$$
$$\leq 1 - \mathbb{H}[A^{\pi}|Z] \,/\, \mathbb{H}[A^{\pi}] + \mathbb{H}[A^{\pi}|Z] \,/\, \mathbb{H}[A^{\pi}] \tag{9}$$
$$= 1$$

where (7) is by definitions of $E^{\pi}$ and $C^{\pi}$, (8) holds since $Z \perp\!\!\!\perp A^{\pi}|X$ according to our problem setup, and (9) is because conditioning never increases entropy.  □

**Proof of Proposition 2**    This proposition is a corollary of Proposition 1. If policy $\pi$ is a perfect expert such that $E^{\pi} = 1$, then $E^{\pi} + C^{\pi} \leq 1$ implies that $C^{\pi} = 0$ (since $C^{\pi} \in [0,1]$).  □

**Proof of Proposition 3**    We have

$$E_{pred}^{\pi} = 1 - \mathbb{H}[A^{\pi}|Y_1 - Y_0] \,/\, \mathbb{H}[A^{\pi}]$$
$$\leq 1 - \mathbb{H}[A^{\pi}|Y_0, Y_1, Y_1 - Y_0] \,/\, \mathbb{H}[A^{\pi}] \tag{10}$$
$$= 1 - \mathbb{H}[A^{\pi}|Y_0, Y_1] \,/\, \mathbb{H}[A^{\pi}] \tag{11}$$
$$= E_{prog}^{\pi}$$

where (10) is because conditioning never increases entropy, and (11) holds since $Y_1 - Y_0$ is a deterministic function of $(Y_0, Y_1)$ hence $Y_1 - Y_0 \perp\!\!\!\perp A^{\pi}|Y_0, Y_1$.  □

**Proof of Proposition 4**    If $C^{\pi} = \mathbb{H}[A^{\pi}|X]/\mathbb{H}[A^{\pi}] = 0$, then

$$\mathbb{H}[A^{\pi}|X] = -\sum_{x \in \mathcal{X}, a \in \{0,1\}} \mathbb{P}\{A^{\pi} = a, X = x\} \log \mathbb{P}\{A^{\pi} = a|X = x\}$$
$$= \sum_{x \in \mathcal{X}, a \in \{0,1\}} \alpha[x] \cdot (-\pi(x)[a] \log \pi(x)[a]) \tag{12}$$
$$= 0$$

First, notice that both $\alpha[x]$ and $-\pi(x)[a] \log \pi(x)[a]$ are non-negative for all $x \in \mathcal{X}, a \in \{0,1\}$, hence each summand in (12) must be equal to zero (for the overall sum to also be equal to zero). Now, consider some $x^* \in \mathcal{X}$ such that $\alpha[x^*] > 0$. This would imply that $-\pi(x^*)[a] \log \pi(x^*)[a] = 0$ for all $a \in \{0,1\}$, which in turn would imply that $\pi(x^*)[a] = 0$ or $\pi(x^*)[a] = 1$ for all $a \in \{0,1\}$. Since it cannot be the case that $\pi(x^*)[0] = \pi(x^*)[1] = 1$, there exists some $a^* \in \{0,1\}$ such that $\pi(x^*)[a^*] = 0$.  □

## D  ADDITIONAL EXPERIMENTS

In the main paper, we focused on a single simulation environment based on the covariate from the TCGA dataset (Weinstein et al., 2013; Schwab et al., 2020). In this section, we repeat our experiments for two more environments: (i) *Linear News*, which is based on the covariates from the News dataset (Newman, 2008), and (ii) *Non-linear TCGA*, which is still based on the TCGA dataset, but in which, outcomes have a non-linear relationship to features. In *Non-linear TCGA*, outcomes are generated as

$$Y_a = e^{-\langle w_{prog}, x_{prog}\rangle^2/2} + e^{-\langle w_a, x_{pred}\rangle^2/2} + \eta_a \tag{13}$$

The results for *Linear News* are given in Figures 7, 8, and Table 2, while the results for *Non-linear TCGA* are given in Figures 9, 10, and Table 3. These results mostly support the same conclusions as the main results in Figures 4, 5, and Table 1. One notable exception is that, while *Expertise-informed* still achieves the best-of-both-worlds performance in *Linear News*, it fails to do so in *Non-linear TCGA*. This is because, in *Non-linear TCGA*, *Balancing* no longer happens to be the best performing method under prognostic expertise, which *Expertise-informed* relies on. Otherwise, *Expertise-informed* still correctly identifies the type of expertise present in datasets as it still matches the performance of *Action-predictive* under predictive expertise and of *Balancing* under prognostic expertise. Regarding why *Balancing* does not perform as well in *Non-linear TCGA*, this might be a side effect of the fact that we implemented all algorithms specifically linear outcomes in mind (see Appendix F for details).

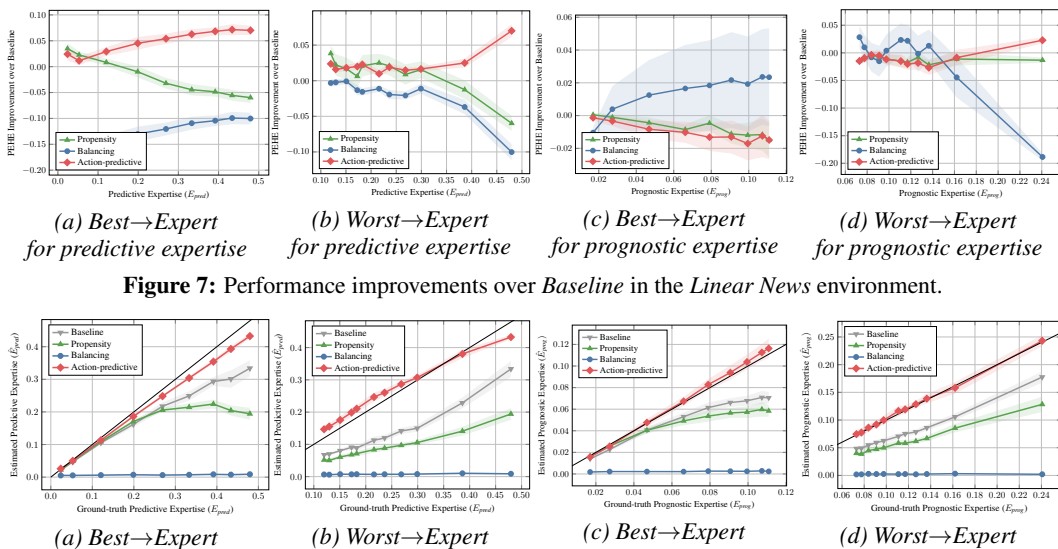

*(a) Best→Expert for predictive expertise*    *(b) Worst→Expert for predictive expertise*    *(c) Best→Expert for prognostic expertise*    *(d) Worst→Expert for prognostic expertise*

**Figure 7:** Performance improvements over *Baseline* in the *Linear News* environment.

*(a) Best→Expert for predictive expertise*    *(b) Worst→Expert for predictive expertise*    *(c) Best→Expert for prognostic expertise*    *(d) Worst→Expert for prognostic expertise*

**Figure 8:** Estimated predictive/prognostic expertise in the *Linear News* environment.

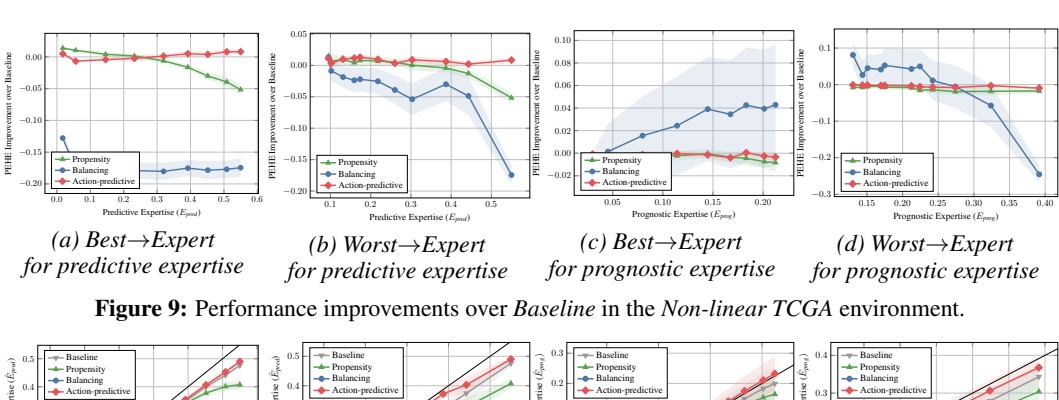

*(a) Best→Expert for predictive expertise*    *(b) Worst→Expert for predictive expertise*    *(c) Best→Expert for prognostic expertise*    *(d) Worst→Expert for prognostic expertise*

**Figure 9:** Performance improvements over *Baseline* in the *Non-linear TCGA* environment.

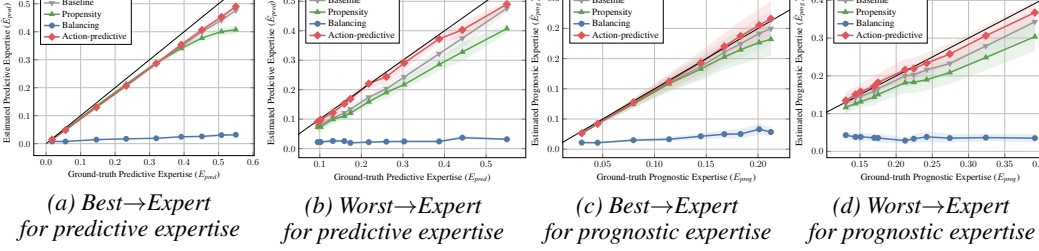

*(a) Best→Expert for predictive expertise*    *(b) Worst→Expert for predictive expertise*    *(c) Best→Expert for prognostic expertise*    *(d) Worst→Expert for prognostic expertise*

**Figure 10:** Estimated predictive/prognostic expertise in the *Non-linear TCGA* environment.

**Table 2:** PEHE of various methods averaged across datasets from the *Linear News* environment.

| Method | Predictive Datasets | Prognostic Datasets | All Datasets |
|---|---|---|---|
| Baseline | 0.495 (0.082) | 0.847 (0.155) | 0.671 (0.118) |
| Propensity | 0.786 (0.126) | 1.511 (0.259) | 1.149 (0.193) |
| Balancing | 0.556 (0.083) | **0.832 (0.148)** | 0.694 (0.115) |
| Action-predictive | **0.465 (0.085)** | 0.839 (0.154) | 0.652 (0.119) |
| Expertise-informed | **0.465 (0.085)** | **0.832 (0.148)** | **0.648 (0.117)** |

**Table 3:** PEHE of various methods averaged across datasets from the *Non-linear TCGA* environment.

| Method | Predictive Datasets | Prognostic Datasets | All Datasets |
|---|---|---|---|
| Baseline | 0.403 (0.069) | **0.305 (0.055)** | 0.354 (0.062) |
| Propensity | 0.408 (0.069) | 0.315 (0.057) | 0.361 (0.063) |
| Balancing | 0.496 (0.070) | 0.320 (0.055) | 0.408 (0.063) |
| Action-predictive | **0.392 (0.069)** | **0.305 (0.057)** | **0.349 (0.063)** |
| Expertise-informed | **0.392 (0.069)** | 0.320 (0.055) | 0.356 (0.062) |

# E  DETAILS OF THE SIMULATION ENVIRONMENT

In the environments based on the TCGA dataset, we used the measurements from the 100 most variable genes—these are all continuous features and the data is log-normalized with each feature scaled in the $[0, 1]$ interval. Meanwhile, the News dataset consists of 10,000 randomly samples news items,each with 2858 word counts. Similar to Shalit et al. (2017), we perform principal component analysis (PCA) and use the first 100 principal continuous features as covariates. In the case of both dataset, we decompose the feature space as $(X_{prog}, X_{pred}, X_{irr}) \in \mathbb{R}^{40 \times 40 \times 20}$—that is $d_{prog} = d_{pred} = 40$ and $d_{irr} = 20$. During this decomposition, we choose which features are prognostic, predictive, or irrelevant completely at random.

We consider a variety of policies in our environments, namely $\pi_{soft}$ and $\pi_{mis}$ as the predictive settings, and $\pi_{non\text{-}pred}$ as the prognostic settings. In the predictive setting, for *Best→Expert*, we fix $d = 0$ and vary $\beta \in \{0.25, 0.28, 0.33, 0.40, 0.50, 0.66, 1, 2, 10\}$, and for *Worst→Expert*, we fix $\beta = 0.25$ and vary $d \in \{0, 2, 4, \ldots, d_{irr} = 20\}$ (Figure 2). In the prognostic setting, for *Best→Expert*, we fix $d = d_{irr}/2 = 10$ and vary $\beta \in \{0.25, 0.28, 0.33, 0.40, 0.50, 0.66, 1, 2, 10\}$, and for *Worst→Expert*, we fix $\beta = 0.25$ and vary $d \in \{0, 2, 4, \ldots, d_{irr} = 20\}$.

# F  DETAILS OF THE BENCHMARK ALGORITHMS

We benchmark across four models: TARNet (i.e. *Baseline*), TARNet with IPW (i.e. *Propensity*), CFRNet (i.e. *Balancing*), and DragonNet (i.e. *Action-predictive*). We use the PyTorch implementations of these models provided in the python package CATENets (Curth & van der Schaar, 2021a). All models have a representation network (i.e. function $\phi$) with one hidden layer with 100 hidden units, and linear prediction layers (i.e. function $f$) for potential outcomes as well as action predictions when applicable. We use dense layers with the ReLU activation function. All models are trained using the Adam optimizer with learning rate 0.001, batch size 1024, and early stopping on a validation set, where we employ a standard train-validation split of 70%–30%. We used a virtual machine with six 6-Core Intel Xeon E5-2690 v4 CPUs, one Tesla V100, and 110GB of RAM to run all experiments. For all experiment results, we average over 10 seeds to obtain error bars.

Computing the ground-truth expertise as well as estimating it using any one of our benchmark algorithms would normally require knowing the marginal distributions of potential outcomes $Y_0, Y_1$. Since we do not have access to these marginal distributions, we rely on a numerical approximation instead. Whenever we need to compute an expertise measure for a dataset $\mathcal{D}$, we first build a histogram of the observed outcomes $\{y^i\} \in \mathcal{D}$ by letting numpy automatically determine which bins to use: $\mathcal{Y} = \mathcal{Y}_1 \cup \cdots \cup \mathcal{Y}_k$. Then, we discretize outcome predictions/observations $\{y_0^i\}$ and $\{y_1^i\}$ according to those bins, which leaves us with an approximate categorical distribution of potential outcomes $Y_0, Y_1$ to compute the expertise with as follows:

$$\hat{E}_{prog} = 1 - \frac{\sum_{a \in \{0,1\}, j_0, j_1 \in [k]} \frac{|i:a^i=a, y_0^i \in \mathcal{Y}_{j_0}, y_1^i \in \mathcal{Y}_{j_1}|}{n} \log_2 \frac{|i:a^i=a, y_0^i \in \mathcal{Y}_{j_0}, y_1^i \in \mathcal{Y}_{j_1}|}{|i:y_0^i \in \mathcal{Y}_{j_0}, y_1^i \in \mathcal{Y}_{j_1}|}}{\sum_{a \in \{0,1\}} \frac{|i:a^i=a|}{n} \log_2 \frac{|i:a^i=a|}{n}} \quad (14)$$

$$\hat{E}_{pred} = 1 - \frac{\sum_{a \in \{0,1\}, j \in [k]} \frac{|i:a^i=a, y_1^i - y_0^i \in \mathcal{Y}_j|}{n} \log_2 \frac{|i:a^i=a, y_1^i - y_0^i \in \mathcal{Y}_j|}{|i:y_1^i - y_0^i \in \mathcal{Y}_j|}}{\sum_{a \in \{0,1\}} \frac{|i:a^i=a|}{n} \log_2 \frac{|i:a^i=a|}{n}} \quad (15)$$

