# OpenReview forum: "Defining Expertise: Applications to Treatment Effect Estimation"
_ICLR.cc/2024/Conference — ICLR 2024 poster_

### Official Review · Reviewer_uYNJ · 2023-10-30

**Soundness:** 4 excellent
**Presentation:** 3 good
**Contribution:** 3 good
**Rating:** 8
**Confidence:** 4

**Summary:**

This paper studies the role of expertise in treatment effect estimation. They formalize predictive (actions dependent on treatment effect) and prognostic expertise (actions dependent on potential outcomes) and argue that knowledge of the type of expertise provides a useful inductive bias for selection of a treatment effect estimation method. They further show that the type of expertise present in a dataset can be estimated observationally with a plug-in method, suggesting a meta-algorithm whereby the type of expertise is estimated and the more appropriate method chosen. Experiments are conducted with sweeps over the amount of each type of expertise in the data. The results have implications particularly for the appropriateness of methods that learn balancing representations, as these are shown to worse than other methods in cases with predictive expertise (as the balancing removes information about the treatment effect from the representation) and better than other methods in cases with prognostic expertise (as the shifts present are unrelated to the treatment effect).

**Strengths:**

* This work offers a novel perspective and analytic framework for reasoning about expertise in treatment effect estimation problems. I found it particularly insightful in the context of reasoning about the causes of covariate shift and overlap violations across treatment strata and the implications for estimation.
* The paper is well-written and clear, with high-quality visualizations.
* The experiments are well-designed and convincing in supporting the theoretical claims with empirical evidence.

**Weaknesses:**

The paper takes a somewhat narrow lens with respect to the breadth of relevant prior work considered. It largely focuses on a particular lineage of treatment effect estimation methods that use neural networks for representation learning, balancing, and fitting potential outcome models. The work would be stronger if it could be contextualized better in the full landscape of work on causal inference for treatment effect estimation across fields, including statistics, epidemiology, and econometrics.

It would be particularly prudent to try to understand how this work fits into the broader landscape of approaches that aim to learn doubly-robust estimators that only require one of either the treatment or outcome models correctly and do not require any notion of balancing beyond overlap. For example, see the Augmented Inverse Propensity Weighted Estimator (e.g., Glynn and Quinn 2010) and Targeted Maximum Likelihood Estimation (Schuler and Rose 2017)

References
* Glynn, Adam N., and Kevin M. Quinn. "An introduction to the augmented inverse propensity weighted estimator." Political analysis 18.1 (2010): 36-56.
* Schuler, Megan S., and Sherri Rose. "Targeted maximum likelihood estimation for causal inference in observational studies." American journal of epidemiology 185.1 (2017): 65-73.

**Questions:**

* As a suggestion, it would interesting to understand the relationship between the issues identified with balancing representations for treatment effect estimation in the presence of predictive expertise with the consistency issues identified with balanced representations for domain adaptation (see Johansson 2019 https://proceedings.mlr.press/v89/johansson19a.html).
* Please elaborate on the relationship the expertise measures and mutual information and the motivation for defining new measures given the relationship between them.
* I don’t follow the argument in section 3.1 regarding the relationship between the two types of expertise. It seems odd to say that predictive expertise implies prognostic expertise, because a policy with predictive expertise (dependent only on the treatment effect) would be insensitive to a constant shift in the potential outcomes while a policy with prognostic expertise would be sensitive to it.

---

> ### Author Response · Authors · 2023-11-16
> **Response to Reviewer uYNJ (Part 1/2)**
>
> Dear Reviewer uYNJ, thank you for your thoughtful comments and suggestions! Below, we aim to address all the individual points raised in your review and hope that this alleviates any remaining concerns — please also see the revised paper for changes (highlighted in blue).
>
> ---
>
> **(W1) Specific focus on balancing representations**
>
> We chose to focus on this specific strand of literature because it has emerged as *the predominant way* of tackling confounding biases in the heterogeneous treatment effect estimation problem in machine learning over the last years — balancing representations have appeared in a majority of the publications on heterogeneous treatment effect estimation in machine learning conferences (e.g. Johansson et al., ICML 2016; Shalit et al., ICML 2017; Yao et al., NeurIPS 2018; Hassanpour & Greiner, ICLR 2020; Du et al., 2021; Assaad et al., AISTATS 2021; Huan et al., 20222), and also been extended to other settings, such as survival data (e.g. Chapfuwa et al., CHIL 2021; Curth et al., NeurIPS 2021) and time-series data (e.g. Bica et al., ICLR 2020; Melnychuk et al., ICML 2022; Seedat et al., ICML 2022). Given the widespread use of this technique in the machine learning literature, we believe a focus on balancing representations to be warranted for the audience targeted by this venue.
>
> ---
>
> **(W2) Relationship to doubly robust estimators**
>
> Our work is largely orthogonal to the work on doubly robust estimators from the (bio)statistics and econometrics literature. In particular, the majority of approaches considered in this literature, and in particular also those cited in the review, target the *average treatment effect*, while we study conditional average treatment effects (CATE, i.e. personalized effects, instead of population-average effects). We focus on estimators based on neural networks because these have been the focus of study in the machine learning literature on CATE estimation since the seminal papers by Shalit, Johansson, and Sontag on the topic in 2016/2017.
>
> Nonetheless, some doubly robust approaches have recently been adapted to the CATE estimation setting, for instance giving rise to DR-learner in Kennedy EH, “Towards optimal doubly robust estimation of heterogeneous causal effects,” *arXiv preprint arXiv:2004.14497*, 2020. The approaches considered in our paper could be used therein as plug-ins for the outcomes, where improved outcome estimation by making use of expertise would likely improve estimation of CATEs by the DR-learner downstream — for instance, Curth & van der Schaar (2021a,b) showed experimentally that using neural networks with better architectures to estimate the outcome portion of the DR-learner improves the performance of the DR-learner as a whole.

---

> ### Author Response · Authors · 2023-11-16
> **Response to Reviewer uYNJ (Part 2/2)**
>
> ---
>
> **(Q1) Consistency issues with balanced representations for domain adaptation**
>
> Thank you for highlighting this interesting connection. The issues arising with balancing representations for treatment effect estimation in the presence of predictive expertise are indeed due to *information loss* in the representation as analyzed theoretically in Johansson et al. (2019).
>
> **Changes:** We have now added a footnote in Section 4.1 to highlight that their results could be adapted from their domain adaptation context to our treatment effect estimation context to provide generalization bounds for this setting.
>
> ---
>
> **(Q2) Relationship between expertise and mutual information**
>
> Our definitions of expertise are exactly equal to the mutual information between actions and outcomes. For predictive expertise, $E^{\pi}\_{\textit{pred}}=I(A^{\pi};Y\_1-Y\_0)/\mathbb{H}[A^{\pi}]$, and for prognostic expertise, $E^{\pi}\_{\textit{prog}}=I(A^{\pi};Y\_0,Y\_1)/\mathbb{H}[A^{\pi}]$. We chose to express expertise in terms of conditional entropies rather than in terms of mutual information because we believe that the former type of expression makes it easier to understand why these definitions are intuitive and why they capture mathematically what we might describe as “expertise” in language.
>
> **Changes:** We now highlight that our definitions can equivalently be expressed in terms of mutual information in a brand-new appendix (Appendix A).
>
> ---
>
> **(Q3) Why does predictive expertise imply prognostic expertise?**
>
> Predictive expertise implies prognostic expertise because $Y_1-Y_0$ is a deterministic function of $(Y_0,Y_1)$. If a policy depends on the treatment effect $Y_1-Y_0$ and hence has predictive expertise, we can also say that the same policy depends on the pair $(Y_0,Y_1)$ and hence has prognostic expertise. We can not say the converse — for instance, a policy that only depends on $Y_0$ would have high prognostic expertise, but since it does not depend on $Y_1-Y_0$ directly, it would generally have low predictive expertise.
>
> Formally, the fact that $Y_1-Y_0$ is a deterministic function of $(Y_0,Y_1)$ means that it is generally less informative than $(Y_0,Y_1)$ and at most as informative as $(Y_0,Y_1)$ — knowing the two potential outcomes $Y_0,Y_1$, we can easily compute the treatment effect $Y_1-Y_0$, but not the other way around. We have $\mathbb{H}[Y_1-Y_0|Y_0,Y_1]=0$ therefore  $H[A^{\pi}|Y_1-Y_0]\geq H[A^{\pi}|Y_0,Y_1]$ and therefore $E^{\pi}\_{\textit{pred}}\leq E^{\pi}\_{\textit{prog}}$. Since predictive expertise is a lower bound for prognostic expertise, having predictive expertise would also imply having prognostic expertise.
>
> The example of a constant shift in the potential outcomes is not particularly helpful in understanding this relationship. In the case of a policy that depends only on the treatment effect, the treatment assignments would not be affected by such a constant shift. As long as the treatment assignments remain the same, neither predictive expertise nor prognostic expertise would be sensitive to the constant shift — that is the policy would still have the same predictive expertise and prognostic expertise under any constant shift. This is a desirable property of expertise: Expertise should capture how differences in outcomes from one subject to another impact treatment assignments for those subjects *independent* of the unit/scaling and the zero-point/mean of outcomes.
>
> **Changes:** We have now stated the fact that predictive expertise is a lower bound for prognostic expertise (hence having predictive expertise implies having prognostic expertise) as Proposition 3 with an accompanying proof (see new Appendix B).

---

> > ### Comment · Reviewer_uYNJ · 2023-11-20
> >
> > Thank you for the response and the revisions. I will keep my overall score the same (8).
> >
> > And thank you for the explanation regarding why predictive expertise implies prognostic expertise. The bound framing is helpful.

---

> > > ### Author Response · Authors · 2023-11-20
> > > **Re: Official Comment by Reviewer uYNJ**
> > >
> > > Many thanks for engaging with our response and for the confirmation of your final score. We are glad that you have found the newly-added Proposition 3 helpful. Your comments were invaluable in improving our paper!

---

### Official Review · Reviewer_obyv · 2023-11-01

**Soundness:** 2 fair
**Presentation:** 4 excellent
**Contribution:** 3 good
**Rating:** 8
**Confidence:** 3

**Summary:**

This paper introduces the idea that selection bias due to expertise might be informative in designing and selecting methods for the conditional treatment effect (CATE) estimation.

They differentiate between two types of expertises, which are formally defined using the concept of entropy for actions induced by a certain policy. On the one hand, they define *prognostic expertise* where actions are based on all potential outcomes. It is given as one minus the relation of the entropy conditional on the potential outcomes and the total entropy. On the other hand, *predictive expertise* refers to the case where actions are based on the treatment effects, and is defined analogously as one minus the relation of the entropy conditional on the treatment effect and the total entropy. Note that these definition bound the expertise between zero and one with zero being no expertise and one being perfect expertise.

They draw connections from expertise to the validity of the overlap assumptions. In particular, the higher the expertise of a policy, the lower is the overlap.

They perform multiple experiments using semi-synthetic data and give intuitive explanations for the different performance of state-of-the-art CATE estimation methods.

Based on the theoretical definition of expertise, they propose an estimator for the predictive and prognostic expertise and a pipeline, called "expertise-informed" to that automatically chooses a suitable CATE estimator based on the dominant expertise type.

**Strengths:**

1. The paper seems to be novel in introducing the idea that a specific type of selection bias, namely allocation done by experts, is informative.

2. The paper is very well-written. It provides a great intuition why the idea of formally introducing expertise is fundamental and how it is related to other quantities of interest such as optimality or overlap assumption.

3. The proofs are very detailed and easy to follow.

**Weaknesses:**

*Definition of Expertise*

1. The relationship between in-context action variability and the overlap assumption is just intuitively explained. A formal proof for this relationship is missing.

2. In Figure 1 the axis tick values are missing. (The ticks are also confusing as for $C^{\pi}=0$, the expertise is somewhere between ticks.)

*Application*

3. The authors explain in great detail why some CATE estimator perform intuitively better under some type of expertise than others. A formal proof for this intuition is however missing.

*Estimating Expertise*

4. It is unclear in the proposed pipeline whether the expertise is estimated on same samples used for training or on some hold-out set and what the consequences of this choice have on the outcomes.

**Questions:**

*Definition of Expertise*

1. The entropy is defined with log to base 2. Do you decide on this because the actions are binary, i.e. $A\in$ { 0,1}?

2. Could you formally prove that $C^{\pi} =0$ implies the violation of the overlap assumption?

3. The comment (i) after Proposition 2 is unclear to me. Could you provide an example for that?

*Application*

4. Is it possible to give more formal arguments how the discussed CATE estimators (TARNet, IPW, CFRNet, DragonNet) are related to the entropy definition of expertise?

5. Are the results sensitive to the choice of network based CATE estimators in the analysis?

*Estimating Expertise*

6. Could you please provide a more detailed description of your pipeline process, in particular whether the same samples are used for estimating expertise and treatment effect? Do you anticipate pre-testing problems in this case?

7. How sensitive are the outcomes of the pipeline to the threshold 1\2?

---

> ### Author Response · Authors · 2023-11-16
> **Response to Reviewer obyv (Part 1/2)**
>
> Dear Reviewer obyv, thank you for your thoughtful comments and suggestions! Below, we aim to address all the individual points raised in your review and hope that this alleviates any remaining concerns — please also see the revised paper for changes (highlighted in blue).
>
> ---
>
> **(Q1)** *The entropy is defined with log to base 2. Do you decide on this because the actions are binary, i.e. $A^{\pi}\in\{0,1\}$?*
>
> Yes, indeed the entropy is defined with $\log_2$ because the actions are binary. This is a fairly arbitrary decision. When $\log_2$ is used, the variability of a uniformly random policy happens to be one.
>
> ---
>
> **(Q2)** *Could you formally prove that $C^{\pi}=0$ implies the violation of the overlap assumption?*
>
> **(W1)** *The relationship between in-context action variability and the overlap assumption is just intuitively explained. A formal proof for this relationship is missing.*
>
> Yes, it is possible to formally prove this relationship. Thank you for the suggestion.
>
> **Changes:** We have now stated the fact that having zero in-context action variability implies that the overlap assumption is violated as Proposition 4 with an accompanying proof (see new Appendix B).
>
> ---
>
> **(Q3)** *The comment (i) after Proposition 2 is unclear to me. Could you provide an example for that?*
>
> An example can be found in the paragraph titled “How does expertise differ from optimality?” in Section 3.1. There, we considered $\mathbb{E}[Y^{\pi}]$ to be our success measure. While $\pi_{\textit{risk}}$ was suboptimal with respect to this success measure, it happened to be a perfect predictive expert. This is because $\pi_{\textit{risk}}$ was aiming to achieve a different objective: making risk-averse decisions.
>
> ---
>
> **(W2)** *In Figure 1 the axis tick values are missing. (The ticks are also confusing as for $C^{\pi}=0$, the expertise is somewhere between ticks.)*
>
> **Changes:** We have now added tick labels to Figure 1. However, the point of Figure 1 is more conceptual than quantitative, the important information is the relative positions of the three regions (smaller/larger expertise, smaller/larger in-context action variability).
>
> ---
>
> **(Q4)** *Is it possible to give more formal arguments how the discussed CATE estimators (TARNet, IPW, CFRNet, DragonNet) are related to the entropy definition of expertise?*
>
> **(W3)** *The authors explain in great detail why some CATE estimator perform intuitively better under some type of expertise than others. A formal proof for this intuition is however missing.*
>
> We believe that a theoretical analysis of existing CATE estimators is out of the scope of this initial paper on defining expertise, which not only provides and motivates a formal definition of expertise for the first time, but also theoretically analyzes its relationship to the CATE estimation problem (Propositions 1, 2, and now 3, 4), empirically explores its relationship to the existing CATE estimators, and even highlights model selection as one of its practical use cases.
>
> ---
>
> **(Q5)** *Are the results sensitive to the choice of network based CATE estimators in the analysis?*
>
> We have used the same network architecture for all CATE estimators. This means that any similarities and differences that are observed between the estimators should be the result of how the estimators are trained rather than the capacity/complexity of the network architecture they use.

---

> ### Author Response · Authors · 2023-11-16
> **Reponse to Reviewer obyv (Part 2/2)**
>
> ---
>
> **(Q6)** *Could you please provide a more detailed description of your pipeline process, in particular whether the same samples are used for estimating expertise and treatment effect? Do you anticipate pre-testing problems in this case?*
>
> **(W4)** *It is unclear in the proposed pipeline whether the expertise is estimated on same samples used for training or on some hold-out set and what the consequences of this choice have on the outcomes.*
>
> When practically implementing *Expertise-informed*, given a training dataset, we always train two models: *Action-predictive* and *Balancing* using the same training dataset. We then use the trained *Action-predictive* model to estimate predictive and prognostic expertise, and use those estimates to select which model to use during test time on a held-out testing dataset. This means that *Expertise-informed* never sees the testing dataset until training, selecting, and committing to a particular model for testing. Here, the fact that the same training dataset is used both for model training and model selection should have minimal impact on results because model selection relies on estimating expertise for the entire training dataset while model performance is measured with respect to the accuracy of estimating treatment effects for individual data points. It would have been a different story if model selection also relied on performance in estimating treatment effects (the same task we want our models to achieve).
>
> ---
>
> **(Q7)** *How sensitive are the outcomes of the pipeline to the threshold 1\2?*
>
> Varying the threshold used in *Expertise-informed* would shift its performance between that of *Balancing* and *Action-predictive*. As the threshold approaches 0, we expect the performance of *Expertise-informed* to converge to that of *Balancing* (as all datasets would be classified as prognostic), and as the threshold approaches 1, we expect its performance to converge to that of *Action-predictive* (as all datasets would be classified as predictive). An important point to consider is that the performance of *Expertise-informed* would never drop below the minimum performance between *Balancing* and *Action-predictive* as it always selects one of these methods or the other.

---

> > ### Comment · Reviewer_obyv · 2023-11-21
> > **Re: Reponse to Reviewer obyv**
> >
> > Dear authors, thank you for your detailed answers and adding the additional proofs (in particular Proposition 4) in the appendix. They are very insightful and addressed my questions. I'll keep my rating (8: accept, good paper).

---

> > > ### Author Response · Authors · 2023-11-22
> > > **Re: Re: Response to Reviewer obyv**
> > >
> > > Thank you very much for engaging with our response and for the confirmation of your score. We are glad that our answers, as well as the newly-added Proposition 4, were insightful and addressed all your questions. Your comments have been invaluable in improving our paper!

---

### Official Review · Reviewer_ScD8 · 2023-11-06

**Soundness:** 4 excellent
**Presentation:** 2 fair
**Contribution:** 1 poor
**Rating:** 5
**Confidence:** 2

**Summary:**

THis paper extends methods for estimating causal treatment from observational data in the presence of expert actions, e.g. the treatment decisions of a clinician.  There is value in the information available from the expert's actions;  What can we learn about the effectiveness of treatment from their actions as domain experts? As mentioned in the  paper "These methods become susceptible to confounding... and treatments (are assigned) based on factors that infuend the outcomes..." This is related to "confounding by indication" [ Salas, M., Hotman, A., Stricker, B.H.: Confounding by indication: an example of variation in the use of epidemiologic terminology. American journal of epidemiology 149(11), 981–983 (1999)]. Confounding occurs because the expert's action conditions both the treatment applied and the outcome

The paper makes a distinction between "predictive" expertise (knowledge of likely outcome,  specifically of Y_1 - Y_0, as occurs in healthcare) and "prognostic" expertise. (knowledge of potential outcomes, Y_1, Y_0 as occurs in education) Prognostic expertise implies predictive, since Y_1 - Y_0 can be determined by knowlege of Y_1 and Y_0.

For this to work one must take into account "overlap" - the variability due to the decision-maker's imperfect knowledge; equivalently the possibility of perfect expertise.  The paper makes the point that one needs the additional distinction between "predictive" and "prognostic."

**Strengths:**

The paper shows that it is possible to take expertise into account and exploit the value of a decision-maker's inductive bias. It concludes that the type of expertise affects the performance of methods for estimating treatment effect (because such methods rely on the difference Y_1 - Y_0,?) , and that the difference in type of expertise may be evident as a way to distinguish datasets.

**Weaknesses:**

The prognostic - predictive distinction on which the paper depends is so subtle that it is hard to see how it is of any consequence.  This may be just a lack of clarity and familiarity with the current economic literature on causality, but it seems to rely on a strong claim that characteristically clinical treatment decisions would be ignorant of outcomes, but only of the _difference_ in outcomes, a claim whose reasons are obscure.  Both types of expertise are defined in terms of expectations of actions conditioned on outcomes as in Equations (2) and (3) -- definitions that leave this question of the distinction open.

Such formulation of expertise  as '.. to what extent the actions of a decision-maker are informed by what a subject’s potential outcomes" seem to contradict the basic notion that experts "assign treatments based on factors that influence the outcomes." What influences what?  In simple terms, the decision-maker's action depends on their state of information -- the features known to them at the time they make their decision. One presumes that thinking of actions influenced by subsequent outcomes is a convention in the economics literature on causality.  This may be necessary for the methods of analysis but it is far from intuitive.

In summary, the significance of the paper relies on a questionable distinction.

**Questions:**

In this conference venue why not draw upon the ML literature on causality (e.g. Pearl's work) and its use of structural equation models? I presume this touches on a long standing discussion that encompasses much more than just the claims of the paper, but such a comparison would open the field to a larger audience, and possibly, by recourse to additional domain knowledge elucidate why it is so that "Cases where more information can be gained through expertise (i.e. cases with high expertise) happen to align with cases where treatment effect estimation is particularly hard." (But other methods may be able to identify the causal effect.)

---

> ### Author Response · Authors · 2023-11-16
> **Response to Reviewer ScD8**
>
> Dear Reviewer ScD8, thank you for your thoughtful comments and suggestions! Below, we aim to address all the individual points raised in your review and hope that this alleviates any remaining concerns — please also see the revised paper for changes (highlighted in blue).
>
> ---
>
> **(W1) Prognostic-predictive distinction**
>
> We would like to emphasize that the distinction between prognostic and predictive variables is made within the medical and biostatistics literature (e.g. Ballman, 2015; Sechidis et al., 2018) outside of our work. We simply extend the same distinction to policies — that is to say, we differentiate between policies that depend on prognostic variables and policies that depend on predictive variables).
>
> Policies that have predominantly predictive expertise — meaning policies that largely depend on the difference between the two potential outcomes (but not on the potential outcomes individually) — can arise naturally when the goal of a decision-maker is to maximize outcomes. For instance, consider the policy that always selects the action that would lead to the best estimated outcome: $\pi(x)=1$ if $\mathbb{E}[Y_1|X=x]>\mathbb{E}[Y_0|X=x]$ and $\pi(x)=0$ otherwise — it has maximal predictive expertise. Since achieving better outcomes for patients is often the primary goal of clinicians when treatment decisions (e.g. Graham et al., 2007; Caye et al., 2019), such policies can be more prevalent (compared with policies with predominantly prognostic expertise but no predictive expertise) in the healthcare domain.
>
> ---
>
> **(W2) Relationship between features, actions, and outcomes**
>
> In our framework, features cause actions, and actions (together with features) cause outcomes. Features $X$ have a distribution $\alpha$ independent of any policy, actions $A^{\pi}$ are determined on the basis on features only such that $A^{\pi}\sim\pi(X)$, and outcomes are a joint consequence of both: $Y=\rho_{A^{\pi}}(X)$.
>
> When we say “to what extent the actions of a decision-maker are informed by what a subject’s potential outcomes **could be**,” our intention is not to imply that a decision-maker’s actions are caused/influenced by potential outcomes, which would be unknown at the time of decision-making. Of course, we agree that in any practical setting, a decision-maker’s actions can only depend on the information that is available to them — in our formulation, this would be features $X$. Some of these features would cause/influence/determine the potential outcomes $Y_0,Y_1$ (let those features be $X_{\textit{critical}}$) and some would not. According to our definition, an expert would mainly consider features $X_{\textit{critical}}$ over $X\setminus X_{\textit{critical}}$, estimate what $Y_0,Y_1$ **could be** based on these features, and make a decision accordingly. With expertise, we aim to capture to what extent a decision-maker follows this process — meaning to what extent their actions are informed by estimated outcomes $\hat{Y}\_0,\hat{Y}\_1$ based on $X\_{\textit{critical}}$.
>
> **Changes:** Thank you for bringing the possibly confusing phrasing of our intuitive definition of expertise to our attention. We have now rephrased the sentence to make it clear that actions never directly depend on potential outcomes.
>
> ---
>
> **(Q1) Why not use the graphical framework of Pearl?**
>
> We refrained from using the graphical framework of Pearl because it is specifically not well-equipped to distinguish between prognostic and predictive variables. This is because DAGs are notoriously bad at representing effect modification (i.e. predictive variables) natively. There have been proposals to extend the graphical framework to better depict effect modifiers (Weinberg CR, “Can DAGs clarify effect modification?” *Epidemiology 18*, 2007), but to the best of our knowledge, no solution has been well established in the literature to this data.
>
> **Changes:** Thank you for bringing Pearl’s framework to our attention as an alternative formulation. We have now included this discussion in a brand-new appendix (Appendix A).

---

### Official Review · Reviewer_7Wu5 · 2023-11-07

**Soundness:** 2 fair
**Presentation:** 3 good
**Contribution:** 3 good
**Rating:** 6
**Confidence:** 2

**Summary:**

The paper discusses the notion that expert decision-makers can implicitly use their domain knowledge when choosing actions, like prescribing treatments. These actions, in turn, can reveal insights about the domain. For example, frequently prescribed treatments might be more effective. However, current machine learning methods may fail to capitalize on the concept of "expertise" as a guiding factor. Specifically, in the context of estimating treatment effects, the prevailing assumption is simply that treatments overlap without considering expert behavior. The paper proposes that recognizing two types of expertise: (1) predictive and (2) prognostic, possessed by decision-makers may enhance the methodology and selection of models for estimating treatment effects. The authors show that understanding the predominant expertise in a domain may significantly impact the performance of these models and that it is possible to identify the type of expertise in a dataset to inform model choice.

**Strengths:**

The paper presents the following advantages:

Firstly, it effectively communicates its proposed concepts through relatable examples, such as those from the medical and teaching fields, introducing and distinguishing between predictive and prognostic expertise.

Secondly, it addresses an intriguing problem that may often be overlooked in causal inference: the incorporation of domain knowledge. By shedding light on this issue, the paper emphasizes the importance of considering expert behavior in the analysis, which could lead to more accurate and informed causal inferences.

**Weaknesses:**

Please see more details in the Questions section.

**Questions:**

1. The paper proposed two novel concepts to define the expert knowledge: (1) Prognostic expertise: $\mathbb{E}^{\pi}_{prog} = 1 - \mathbb{H}[A^{\pi}|Y_0, Y_1] / \mathbb{H}[A^_{\pi}]$. (2) Predictive expertise: $\mathbb{E}^{\pi}_{pred} = 1 - \mathbb{H}[A^{\pi}|Y_1 - Y_0] / \mathbb{H}[A^_{\pi}]$. The rationale for selecting entropy over other statistical measures, such as the variance of actions, is not immediately clear. Can you elucidate the benefits of employing entropy in this context? What are the specific advantages of entropy in capturing the nuances of expert decision-making, and under what conditions does it outperform simpler measures like variance?

2. The consideration of domain expert knowledge in personalized decision-making problems is crucial but challenging, especially given that such expertise may not always be accurate or easily discernible in real data sets. How might one effectively balance the integration of domain knowledge with data-driven insights to mitigate the risk of relying on potentially inaccurate expertise? Is there a methodology within your framework that utilizes data-driven approaches to validate and calibrate the contributions of domain expertise, particularly through the lens of prognostic and predictive expertise measures?

3. In the context of clinical trials, where treatment assignments are typically randomized and propensity scores are uniformly distributed. Then, the application of expert knowledge is less straightforward. Considering such scenarios, how do the proposed prognostic and predictive expertise metrics maintain their utility? Can these metrics still provide meaningful information for personalized treatment recommendations when the baseline assumption of expert influence on treatment assignment is removed? How might these metrics be adapted or interpreted in a randomized trial environment to enhance personalized medicine approaches?

---

> ### Author Response · Authors · 2023-11-16
> **Response to Reviewer 7Wu5**
>
> Dear Reviewer 7Wu5, thank you for your thoughtful comments and suggestions! Below, we aim to address all the individual points raised in your review and hope that this alleviates any remaining concerns — please also see the revised paper for changes (highlighted in blue).
>
> ---
>
> **(Q1) Why entropy over statistical measures?**
>
> Although variance, similar to entropy, could also be indicative of how “random” a random variable is, the two quantities measure fundamentally different things: Variance quantifies the amount of spread around a mean while entropy quantifies uncertainty. When defining expertise, our aim is to capture the uncertainty of outcomes when the actions of a decision-maker are known vs. not known to an outside observer (the bigger the difference between the two cases, we say the larger the expertise is). Hence, a more direct measure of uncertainty is naturally more suitable to our aim and use case.
>
> On a more technical level, attempting to define expertise through variance has two immediate shortcomings:
>
> 1. First, variance is not well defined for categorical variables, such as binary actions in our work, unless we assign numerical values to each category. Even if we were to assign such numeric values (for instance, $A^{\pi}=0$ for the negative treatment and $A^{\pi}=1$ for the positive treatment), the resulting variance would be sensitive to our assignments (for instance, the variance of $A^{\pi}$ would have increased if we were to represent the negative treatment as $A^{\pi}=-1$ instead of $A^{\pi}=0$, which should not be a meaningful change as far as expertise is concerned).
>
> 2. Variance depends on the scale of variables whereas entropy does not. For instance, the variance of $2Y$ would be double the variance of $Y$, while their entropies would be the same: $\mathbb{H}[Y]=\mathbb{H}[2Y]$. Such sensitivity to scale is undesirable when defining expertise — the fact that outcomes are recorded twice as large in a dataset (maybe due to a change of units) should have no effect on expertise.
>
> **Changes:** Thank you for bringing statistical measures to our attention as an alternative to entropy. We have now included this discussion in a brand-new appendix (Appendix A).
>
> ---
>
> **(Q2) Data-driven approaches to discerning domain expertise**
>
> Our experiments reveal that different methods are more or less suitable to settings with different types of domain expertise. We strongly agree that this makes it extremely important to accurately discern the type of expertise present in a domain or dataset when selecting between different methods. While machine learning traditionally relies on domain knowledge in making such decisions, our framework enables a more data-driven and quantitative approach: We propose the pipeline “expertise-informed” in Section 4.2 precisely to demonstrate one such approach.
>
> Notice that *Expertise-informed* first identifies the prominent type of expertise present in a dataset by estimating the measures of prognostic and predictive expertise (and considering their ratio). Then, it takes advantage of this identification by selecting the most appropriate algorithm for estimating treatment effects: *Action-predictive* for datasets with predominantly predictive expertise vs. *Balancing* for datasets with predominantly prognostic expertise.
>
> ---
>
> **(Q3) Randomized controlled trials**
>
> You are correct that, in a randomized controlled trial, when the propensity scores are uniform across treatments, there would be no expertise — and appropriately, both the predictive and the prognostic expertise would be equal to zero for datasets collected through a randomized controlled trial (this can be inferred from Proposition 1, as we mention in the paragraph above Proposition 1, a uniform policy would have $C=1$ hence $E_{\textit{pred}}=0$ and $E_{\textit{prog}}=0$). Consequently, expertise as an inductive bias would of course be less helpful in estimating treatment effects using such datasets, however in that case, the datasets would already be ideal for treatment effect estimation with no confounding bias (the best case scenario in Figure 1). This is why we mainly focused on the high expertise, low overlap setting (the amber region in Figure 1).
>
> While taking advantage of expertise would not be possible for datasets collected through a randomized controlled trial, estimating expertise (as in Section 4.2) can still act as a data-driven way to determine whether the data we have is effectively randomized or not: The closer both $E_{\textit{pred}}$ and $E_{\textit{prog}}$ are to $0$, the closer the data would be to trial data, in which case we might prefer conventional supervised learning methods over algorithms specialized for treatment effect estimation.
>
> **Changes:** We have now included this discussion in a brand-new appendix (Appendix A).

---

### Author Response · Authors · 2023-11-16
**Global Response**

Dear Reviewers,

Thank you for your thoughtful comments and suggestions!

We are encouraged that you found our framework **novel** for “introducing the idea that a specific type of selection bias, i.e. allocation done by experts, is informative” (*obyv*), **intriguing** for considering “a problem that may often be overlooked in causal inference: the incorporation of domain knowledge” (*7Wu5*), and **insightful** when “reasoning about the causes of covariate shift and overlap violations” (*uYNJ*).

We are particularly pleased that our presentation has been described as “effective with relatable examples” (*7Wu5*), “well-written and clear” (*obyv*, *uYNJ*) with “detailed and easy to follow proofs” (*obyv*) and “high-quality visualizations” (*uYNJ*). Moreover, our experiments were “well-designed and convincing in supporting the theoretical claims with empirical evidence” (*uYNJ*).

We have also taken your feedback into account and made the following key changes to improve our paper (highlighted in blue):

1. Added a brand **new appendix (Appendix A)** for further discussion on expertise, including the topics: (i) Why entropy over statistical measures? (ii) Why not use the graphical framework of Pearl? (iii) Expertise in terms of mutual information, and (iv) Expertise in experimental data,
2. Added another brand **new appendix (Appendix B)** titled “Theory of expertise,” which formally states facts that were previously only stated on an intuitive level as brand **new propositions (Proposition 3 and 4)** with accompanying proofs in Appendix C,
3. Rephrased parts of the main text that were found possibly confusing,
4. Referenced additional related work about potential connections of balancing representation in our context to balancing representations in domain adaptation.

We hope these updates address any remaining concerns — we are more than happy to respond to any additional concerns that might arise during the rebuttal period!

Best regards,

The authors of #6457

---

### Comment · Area_Chair_eKxL · 2023-11-20
**reviewers, please acknowledge the responses from the authors**

Dear reviewers: Please read the replies from the authors carefully, and submit your reactions. Please be open-minded in deciding whether to change your scores for the submission, taking into account the explanations and additional results provided by the authors.

Thank you!

---

> ### Author Response · Authors · 2023-11-20
> **We sincerely thank the AC for the message**
>
> Many thanks to the AC for the message. The reviewer’s comments have helped us a great deal in improving the paper, and we hope the reviewers will benefit from the discussion as much as we did. We look forward to hearing the reviewers’ replies.

---

### Meta-Review · Area_Chair_eKxL · 2023-12-15

**Metareview:**

Three reviewers are positive about this submission, while one is slightly negative. Reviews are of good quality.

The paper has two main contributions. One is to discuss the idea of expertise and two types of expertise. The other is to do simulations showing that the best method of estimating personalized treatment effect differs depending on the type of expertise present in data (end of Section 4.1). Both contributions are solid, but have room for improvement.

An improvement for the first contribution could be to change terminology. A standard dictionary definition of "prognostic" is "serving to predict the likely outcome of a disease" so essentially a synonym for "predictive". Based on Section 3.1, "predictive expertise" could be called "effectiveness expertise" while "prognostic expertise" could be called "outcome expertise."

The statement "predictive expertise implies prognostic expertise" is confusing, because (as the authors say) prognostic expertise is stronger than predictive expertise. It might be more clear to just use the words "stronger" and "weaker." The statement "predictive expertise implies prognostic expertise" can be clarified as "non-zero predictive expertise implies non-zero prognostic expertise."

The second contribution is interesting, but there is no direct guidance for practitioners, and no result presented for a real-world causal research question.

Here are two issues not raised by the reviewers.

First, the introduction has an example of predictive expertise in medicine, versus prognostic expertise in education. However, the real issue here is that each patient can receive a different treatment, but all pupils in a class must receive the same shared treatment. It is not obvious that teachers and physicians really have different types of expertise. Instead, they must convert their predictive and/or prognostic knowledge into treatment assignments in very different ways.

Second, the paper should discuss the contrast between predicting an outcome (causal or not) and selecting an action. The latter depends not just on outcomes and benefits, but also on costs. Can costs always be folded into outcomes, as merely negative benefits? In medicine, costs are much easier to predict than outcomes, whether absolute (i.e., "prognostic") or relative (i.e., "predictive').

**Justification For Why Not Higher Score:**

Room for improvement.

**Justification For Why Not Lower Score:**

Reviewers are positive and the contributions are real.

---

### Decision · Program_Chairs · 2024-01-16

Accept (poster)